# MVP: Multi-scale Visual Prompt for Visual AutoRegressive Generation

## Abstract

Prompt tuning, especially perturbation-based prompt tuning, encounters obstacles in visual generation. On the one hand, the autoregressive paradigm, which provides the most ideal environment for prompt tuning, struggles to model planar concept: traditional autoregressive methods employ raster-scan for image modeling, disrupting the spatial structure of images. On the other hand, perturbation-based prompts work as learnable perturbations in pixel space, and their effectiveness comes at quite a little computational cost, making it difficult to balance performance and efficiency. To address these challenges, we propose Multi-scale Visual Prompt (MVP), a perturbation-based prompt tuning method tailored for visual autoregressive generation with planar concept and efficient information propagation. MVP builds on Visual AutoRegressive (VAR) models with next-scale prediction for capturing planar concept, and introduces prompt tokens in the outermost token frame at each scale for efficient signal control and information propagation. During training, we use increasingly detailed tuning text to facilitate prompt learning. Moreover, MVP extends VAR's capability for text-to-image generation. Extensive experiments validate the effectiveness of MVP. Code is [available](#).

## 1 Introduction

Prompt tuning (Li & Liang, 2021; Liu et al., 2021; Lester et al., 2021) enables models to perform specific tasks by introducing the learnable prompt, requiring only minimal fine-tuning without any access to model parameters. In generation tasks, particularly in textual generation based on large language models (LLMs), prompt tuning has achieved remarkable success (Hao et al., 2023; Ajwani et al., 2024; Tang et al., 2022). However, its transfer to visual generation has not reached the same success. Although some embedding-based (Gal et al., 2022; Ruiz et al., 2023) and adapter-based (Yeh et al., 2023; Ye et al., 2023) prompt tuning methods show notable results in visual generation, perturbation-based prompt tuning, which offers stronger controllability and is more suitable for high-dimensional tasks, remains almost unexplored. We identify the following two reasons:

*First, traditional autoregressive methods cannot model planar concept.* The autoregressive paradigm inherently provide an ideal environment for perturbation-based prompt tuning, as they can seamlessly incorporate the learnable prompt: prompt and input within autoregressive models share the same structure, enabling prompt and input tokens to be directly concatenated or added without requiring architectural adjustments or additional modules. Moreover, the autoregressive paradigm also ensures full-coverage prompt control: serving as contextual information, prompt participates in attention computations across all layers and heads, and their influence progressively propagates through autoregressive modeling. In contrast to the diffusion paradigm (Sohl-Dickstein et al., 2015; Ho et al., 2020) requiring thousands of conditional inputs, autoregressive minimizes prompt signal attenuation. Therefore, the autoregressive paradigm serves as the foundation for perturbation-based prompt tuning. However, traditional autoregressive modeling converts visual content into a sequence in raster-scan order, limiting the model's ability to capture original spatial adjacency relationships and two-dimensional structural correlations, ultimately leading to outputs lacking global consistency and suboptimal performance compared to other visual generation paradigms. The emergence of Visual AutoRegressive (VAR) (Tian et al., 2024) addresses this deficiency by transforming the autoregressive modeling pattern from next-token prediction into next-scale prediction. In next-scale prediction, each prediction unit contains a scale-specific feature map, which is a set of multiple tokens predicted simultaneously rather than sequentially. This modeling pattern preserves the spa-

tial structural information of visual content and improves semantic continuity across scales, which allows the autoregressive paradigm to possess planar concept. Consequently, perturbation-based prompt tuning should be implemented in the autoregressive paradigm of next-scale prediction.

*Second, perturbation-based prompt tuning for visual tasks requires substantial computational costs.* Semantic information in natural language is relatively concentrated, while it tends to be more sparse and discrete in the image. For the same semantic content, the image token sequence is typically longer than the text token sequence. Excessive tokens rapidly increase computational burden, against the original intention of prompt tuning. Conversely, insufficient prompt tokens result in inadequate full-coverage control for the modeling process, i.e., some non-prompt tokens are unable to obtain effective signal control and sufficient information propagation from the prompt tokens, hindering long-range dependency modeling. This requires an ingenious prompt design that balances efficiency and performance. A feasible solution is to select a subset of tokens at each scale to incorporate the prompt while ensuring that at least three tokens come from different rows or columns. Since three non-collinear points uniquely define a plane, perturbation-based prompt tuning is allowed to possess and model planar concept. Our derivation and calculation reveal that incorporating the prompt into the tokens located in the outermost frame of the feature map enables a more efficient information propagation to other tokens. Therefore, we suppose that the subset of selected tokens lies in the outermost frame at each scale.

In light of the above discussion, we propose Multi-scale Visual Prompt (MVP), a perturbation-based visual prompt tuning method with planar concept and efficient information transmission, to improve the generation quality and expand the task of VAR. We select the tokens located in the outermost square frame at each scale to introduce the learnable prompt (perturbation), ensuring that the prompt possesses planar concept while maintaining a balance between performance and efficiency. MVP introduces the learnable prompt by adding prompt tokens to selected tokens. Regarding the prompt learning strategy, we design three types of tuning text with incrementally richer semantics: class labels, sentences, and captions. Subsequently, we implement prompt learning through contrastive learning between CLIP embeddings of images (obtained through feature inversion from feature maps at some specific scales) and their corresponding tuning texts. Owing to this design, our method also extends VAR's capability for text-to-image generation through MVP.

To evaluate the effectiveness of MVP, we conduct experiments on the two tasks: improving VAR's class-to-image generation quality and expanding VAR's text-to-image generation capability. With a $d$-16 model on ImageNet (Krizhevsky et al., 2017) at $256 \times 256$ resolution, MVP respectively improves FID and IS scores by 4.1% and 9.7% over VAR. Strikingly, compared to VAR-CLIP (Zhang et al., 2024a), MVP only needs **0.54% training GPU hours** to achieve competitive performance.

## 2 RELATED WORK

**Visual Prompt Tuning**    Since prompt tuning proposed in NLP (Lester et al., 2021), it has rapidly entered computer vision field. VPT (Jia et al., 2022) adds learnable tokens to Vision Transformers and beats full fine-tuning on 20 recognition benchmarks, while ViPT (Zhu et al., 2023) inserts prompt tokens layer-wise to boost tracking via richer features. Research (Bahng et al., 2022; Khattak et al., 2023; Wu et al., 2024) utilize prompt tuning to improve cross-modal alignment, facilitating various discriminative tasks. Nevertheless, in generative tasks, the adoption of prompt tuning has been markedly limited. The few existing works (Sohn et al., 2023; Kumari et al., 2023; Mao et al., 2024; Mou et al., 2024; Guo et al., 2024) are embedding-based and adapter-based prompt tuning, while perturbation-based prompt tuning remains largely unexplored in visual generation.

**Visual Autoregressive Generation**    Recent progress in LLMs (Touvron et al., 2023; Wan et al., 2024; Chung et al., 2024) has spurred equally rapid advances in visual autoregressive generation. By treating image synthesis as a token prediction task (Esser et al., 2021b; Van den Oord et al., 2016; Chen et al., 2020; Esser et al., 2021a), early autoregressive methods relied on vector quantization (VQ) (Van Den Oord et al., 2017) to discretize continuous feature maps, but suffered from quantization error and information loss. Subsequent works removed these bottlenecks. For example, MAR (Li et al., 2024) questioned the need for VQ and utilized continuous latent for autoregressive modeling, while Fluid (Fan et al., 2024) introduced continuous tokens for higher fidelity. Llama-Gen (Sun et al., 2024) improved image generation quality by optimizing a high-fidelity and high-

utilization tokenizer. Inspired by residual quantization methods (Lee et al., 2022; Huijben et al., 2024), VAR (Tian et al., 2024) reframes next-token prediction as next-scale prediction, further unleashing the potential of the autoregressive paradigm to generate high-quality visual content. The emergence of autoregressive methods based on next-scale prediction (Zhang et al., 2024a; Tang et al., 2024; Han et al., 2025; Qu et al., 2025) reflects increased optimism about the future development of visual autoregressive generation. Therefore, we develop a simple yet efficient perturbation-based prompt tuning on the VAR family with next-scale prediction to enhance task performance.

# 3 METHODOLOGY

The overall framework of MVP is illustrated in Figure 1. MVP utilize a set of tokens located in the outermost square frame at each scale to introduce the learnable prompt, which is square frame prompt. As the scale increases, the computational cost of square frame prompt grows rapidly. Therefore, we establish a scale threshold beyond which the number of square frame prompt tokens no longer increases with scale increase, presenting as one square frame prompt transforming into four L-shaped prompts at one scale. The prompt tokens are directly added with the input tokens. In training, we adopt a CLIP to encode multiple tuning texts (sentence and caption) and certain feature maps for contrastive learning to enable MVP learning.

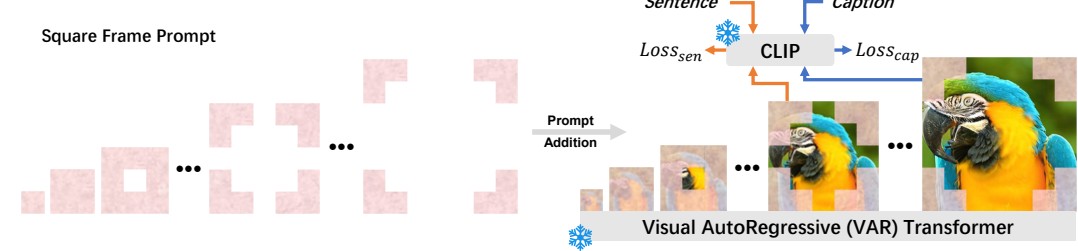

Figure 1: The Overall framework of MVP.

## 3.1 PROMPT DESIGN PRINCIPLE

### 3.1.1 MODEL SELECTION

As discussed in the previous section, it is ideal for MVP to possess planar concept. VAR (Tian et al., 2024) and VAR-like models (Tang et al., 2024; Han et al., 2025) based on next-scale prediction serve as the target models for MVP. By decomposing generation into next-scale residual feature map prediction, VAR naturally introduces two components: an intra-scale residual feature map, representing the spatial plane, and an coarse-to-fine inter-scale sequence, reflecting the temporal progression across scales. The modeling process of VAR involves $T$ multi-scale token feature maps $(R_1, R_2, \cdots, R_T)$ defined by a size set $\{S_1 \times S_1, S_2 \times S_2, \cdots, S_T \times S_T\}$. At the $t$-th scale, VAR predicts the residual feature map $R_t \in \mathbb{R}^{S_t \times S_t}$ based on all previous scales. The autoregressive likelihood can be formulated as follows:

$$p(R_1, R_2, \cdots, R_T) = \prod_{t=1}^{T} p(R_t \mid \langle \text{sos} \rangle, R_1, R_2, \cdots, R_{t-1}), \tag{1}$$

where $(R_1, R_2, \cdots, R_{t-1})$ denotes the "prefix" of $R_t$, and $\langle \text{sos} \rangle$ is the conditional embedding.

### 3.1.2 CENTRAL IMPACT ANALYSIS

After determining the target model, we consider the form of prompt introduction. Research (Zhang et al., 2024b) reveals that although prompt tuning can effectively enhance the performance of models, it may also result in limited performance improvement or significant degradation for other tasks. This is attributed to the fact that the learnable prompt leads to notable changes in the model's visual features, consequently invalidating plenty of knowledge acquired from large-scale pre-training during its transfer to other models, thus impacting their performance. This phenomenon, referred to as model feature corruption, critically impairs overall model performance.

In addition, perturbation-based prompt tuning carries higher feature corruption risks, as it directly introduces perturbation within the pixel space. Therefore, it is essential to employ an appropriate form of prompt introduction to resist image corruptions. In some visual tasks, the methods (Bahng et al., 2022; Wu et al., 2022; Xie et al., 2023) introduce control signals in the pixel space around clean images as a frame, ensuring minimal impact on the image center. The image center typically contains critical information and primary objects. Modifying the image center may cause issues such as subject deformation, semantic drift, and expression distortion, thereby affecting subsequent image-text alignment, understanding, and generation.

Taking the feature map $R_t \in \mathbb{R}^{S_t \times S_t}$ at the $t$-th scale as an example, we divide $R_t$ into $N+1$ concentric square frames by layers. Specifically, Frame 0 is the outermost frame, containing all tokens on the outermost boundary. Frame 1 is the sub-outermost frame, with its tokens positioned just inside Frame 0, forming a second boundary. Following this pattern, Frame N is the center frame, containing the innermost token(s), where $N = \left\lfloor \frac{\min(S_t)}{2} \right\rfloor$. We denote the set of tokens in Frame $n$ as $\mathcal{S}_{t,n}$, for $n = 0, 1, \cdots, N$. Signals (i.e., perturbations) weaken as the propagation distance increases. In maps with hierarchical structure or spatial layout, the impact received by nodes diminishes with increasing distance from the source. Therefore, we define the propagation distance as $dis$ and the signal attenuation factor $\alpha$ related to propagation distance, where the attenuation factor is negatively correlated with propagation distance: $dis \propto -\alpha$. If a perturbation $\delta$ is added to a token in frame $n$, then its impact $I_n$ on the center frame $\mathcal{S}_{t,N}$ is: $\text{Impact}(n \to N) = \delta \cdot \alpha_{N-n}$. Therefore, the impact of introduction from outermost frame 0 and non-outermost frame c on the center frame can be denoted as $I_0 = \delta \cdot \alpha_{N-0}$ and $I_0 = \delta \cdot \alpha_{N-c}$. Since $N - 0 > N - c$, therefore $I_0 < I_c$.

Through both qualitative and quantitative analyses, we demonstrate that introducing prompts in the outermost square frame minimizes impact on the image center, thereby avoiding model feature corruption. Therefore, we propose that MVP incorporates the learnable prompt within the outermost square frame of the feature map at each scale.

### 3.1.3 OVERALL IMPACT ANALYSIS

Introducing the prompt in the outermost square frame ensures minimal impact on the center of images while resisting image corruptions. Meanwhile, the form of token introduction is supposed to achieve good overall propagation efficiency, meaning that prompt tokens are distributed within a reasonable and appropriate proximity to each input token, thereby facilitating input tokens receiving signal control and information propagation from prompt tokens. Therefore, we analyze the overall impact of the outermost square frame prompt introduction across the entire feature map.

For the feature map $R_t \in \mathbb{R}^{S_t \times S_t}$ at the $t$-th scale, we define the index set of all tokens at this scale as $\Omega_t = \{(x,y) \mid x,y \in \{1, 2, \ldots, S_t\}\}$. Therefore, the token index set of the outermost square frame is $\mathcal{B}_t = \{(x,y) \in \Omega_t : (x \in \{1, S_t\}) \lor (y \in \{1, S_t\})\}$. Thus, the minimum distance from any token to the outermost square frame can be represented as follow:

$$dis_{\min}\big((x,y), \mathcal{B}_t\big) = \min_{(u,v) \in \mathcal{B}_t} \|(x,y) - (u,v)\|, \tag{2}$$

where the distance can be measured using Manhattan distance, Euclidean distance, Chebyshev distance, or other suitable metrics.

Given that the linear distances between the given token and the prompt tokens located on the same row or column correspond to the minimum distances to the four outermost boundaries, it follows:

$$dis_{\min}\big((x,y), \mathcal{B}_t\big) = \min\{\, x-1,\ S_t - x,\ y - 1,\ S_t - y \,\}. \tag{3}$$

We define the unified maximum distance $dis_{\max}^{\text{uni}}$ from any token to the outermost square frame as:

$$dis_{\max}^{\text{uni}} := \max_{(x,y) \in \Omega_t} dis_{\min}\big((x,y), \mathcal{B}_t\big). \tag{4}$$

From Equation 3, it can be observed that the token(s) at the geometric center is/are farthest from the outermost square frame, thus $dis_{\max}^{\text{uni}} = \lfloor (S_t - 1)/2 \rfloor$.

Given that $dis_{\max}^{\mathrm{uni}}$ is less than half the scale size $\frac{S_t}{2}$, it can be demonstrated that introducing the prompt in the outermost square frame also achieves a good overall propagation efficiency.

## 3.2 MULTI-SCALE VISUAL PROMPT

Following the prompt design principles described above, MVP introduces learnable prompts in the outermost square frames of feature maps in VAR family models based on next-scale prediction.

### 3.2.1 PROMPT TOKEN SELECTION

For the feature map $R_t \in \mathbb{R}^{S_t \times S_t}$ at the $t$-th scale, MVP selects prompt tokens in the outermost square frame of $R_t$. The number of these prompt tokens at the $t$-th scale is represented as $N^{\mathcal{B}_t}$ and is given by $N^{\mathcal{B}_t} = \mathcal{I}_{\mathrm{id}}^{\mathcal{B}_t} = 4S_t - 4$, where $\mathcal{I}_{\mathrm{id}}^{\mathcal{B}_t}$ is the index set of prompt token positions in the outermost square frame of the feature map $R_t$. Therefore, the prompt token set $\mathcal{V}^{\mathcal{B}_t}$ of the square frame prompt with dimension $D$ can be represented as follow:

$$\mathcal{V}^{\mathcal{B}_t} = [\mathbf{v}_t^1, \ldots, \mathbf{v}_t^{N^{\mathcal{B}_t}}] \in \mathbb{R}^{N^{\mathcal{B}_t} \times D}. \tag{5}$$

In VAR, the number of tokens grows rapidly from small to large scales, which can rapidly increase computational cost, against the original intention of prompt tuning. Although the square frame prompt design has effectively reduced the prompt computational budget from $\mathcal{O}(\mathcal{S}^2)$ to $\mathcal{O}(\mathcal{S})$, the number of square frame prompt tokens still becomes pretty large as the scale increases, requiring more computational cost. Therefore, we set a threshold $\tau$ on the number of square frame prompt tokens to perserve efficiency when the scale is large. Specifically, $\tau$ is the maximum number of square frame prompt tokens. Once the threshold $\tau$ is exceeded, the number of prompt tokens for the scale remains constant. Formally, a square frame prompt is converted into four L-shaped prompts: four corner tokens of the feature map outward along the outermost square frame, incorporating $a$ tokens in every available direction. These combined tokens collectively form the L-shaped prompts. $a = \lfloor (\tau - 4)/8 \rfloor$, thus $N^{\mathcal{B}_t} = 8a + 4$. Update the square frame prompt set $\mathcal{V}^{\mathcal{B}_t} = [\mathbf{v}_t^1, \ldots, \mathbf{v}_t^{N^{\mathcal{B}_t}}]$.

### 3.2.2 PROMPT TOKEN ADDITION

For the $t$-th scale, we utilize the square frame prompt set $\mathcal{V}^{\mathcal{B}_t}$ to construct the prompt feature map $\mathcal{F}_t \in \mathbb{R}^{S_t \times S_t \times D}$ that matches the shape of the feature map $R_t$. All non-prompt positions in $\mathcal{F}_t$ are padded with zeros. Then, we add $\mathcal{F}_{t+1}$ to $R_{t+1}$ to obtain the new feature map $\hat{R}_t$ at the $t + 1$-th scale: $\hat{R}_{t+1} = R_t + \mathcal{F}_{t+1} \in \mathbb{R}^{S_t \times S_t \times D}$. And the autoregressive likelihood can be reformulated as:

$$p(R_1, \ldots, R_T) = \prod_{t=1}^{T} p(R_t \mid \langle \mathrm{sos} \rangle, \hat{R}_1, \ldots, \hat{R}_{t-1}, \mathcal{F}_t). \tag{6}$$

## 3.3 PROMPT LEARNING STRATEGY

Although perturbation-based prompt tuning works in pixel space and can effectively control style, texture, spatial layout, and other visual elements to improve visual generation, they have weaker semantic controllability than embedding-based and adapter-based prompt tuning. Taking embedding-based prompt tuning as an example, it incorporates semantically rich embedding vectors into the feature space, thereby establishing connections with semantic representation. In contrast, perturbation-based prompt tuning essentially introduces learnable perturbation, which exhibits poor interpretability and lacks semantics, resulting in suboptimal semantic expression. Moreover, since VAR is a class-to-image generation model, its class-level conditioning inherently lacks rich semantic information, making the training of perturbation-based prompts significantly challenging.

The above analysis indicates that incorporating richer semantic information is the key to training perturbation-based prompts, allowing prompts to acquire more essential knowledge. It is worth noting that the feature map scales predicted by VAR based on next-scale prediction gradually increase, we suppose that different stages of this coarse-to-fine generation process require different tuning texts. Early stages focus on modeling semantic concept, so class-level text (label) works well for prompt learning. Middle stages refine concept and layout, making sentence-level text suitable. Later

stages enhance details, so caption-level text with more details helps prompts learn richer semantics. Therefore, we propose multi-level semantic refinement as a strategy to improve prompt training.

Specifically, we introduce two more tuning texts: a sentence-level text $\mathcal{T}_{\mathbf{sen}}$ containing relatively comprehensive concepts (using fixed templates such as "a photo of {}"), and a caption-level text $\mathcal{T}_{\mathbf{cap}}$ containing detailed visual attributes and fine-grained semantic information. These tuning texts with difference granularity facilitate MVP to fine-tune VAR, thereby enhancing VAR's semantic expression and generation quality. $K$ is the total number of VAR scales. We set an inter-anchor index $\kappa = \lfloor \beta K \rfloor \in \{1, 2, \ldots, K-1\}$ with a hyper-parameter $\beta \in (0, 1)$ (e.g. $\beta = 0.6$ found by grid search). Based on this anchor, we employ the image $\mathcal{I}_{\mathrm{sen}}$ at the $\kappa$-th scale and the image $\mathcal{I}_{\mathrm{cap}}$ $K$-th scale to enable the prompt to learn from sentence-level and caption-level tuning text, respectively. $\mathcal{I}_{\mathrm{sen}}$ and $\mathcal{I}_{\mathrm{cap}}$ are generated by following processes:

$$\mathcal{I}_{\mathrm{sen}} = \mathrm{Decoder}(\sum_{t=1}^{\kappa} \mathrm{Up}(R_t)), \quad \mathcal{I}_{\mathrm{cap}} = \mathrm{Decoder}(\sum_{t=1}^{K} \mathrm{Up}(R_t)), \tag{7}$$

where $R_t$ denotes the residual feature map predicted at the $t$-th scale, and $\mathrm{Up}(\cdot)$ denotes the up-sample inversion transform function to unify the spatial shapes.

We then apply the CLIP-based loss Radford et al. (2021) to supervise semantic alignment between the generated inversion images and their corresponding tuning texts in a shared embedding space. Following the standard contrastive learning, we define an image-to-text contrastive loss $\mathcal{L}_{\mathcal{IT}}$ and a text-to-image contrastive loss $\mathcal{L}_{\mathcal{TI}}$, and combine them symmetrically as the final CLIP loss $\mathcal{L}_{\mathrm{CLIP}}$. We apply this loss at both the sentence and caption levels, obtaining the total semantic loss:

$$\mathcal{L}_{\mathrm{semantic}} = \lambda_{\mathrm{sen}} \mathcal{L}_{\mathrm{CLIP}}(\mathcal{I}_{\mathrm{sen}}, \mathcal{T}_{\mathrm{sen}}) + \lambda_{\mathrm{cap}} \mathcal{L}_{\mathrm{CLIP}}(\mathcal{I}_{\mathrm{cap}}, \mathcal{T}_{\mathrm{cap}}), \tag{8}$$

where $\lambda_{\mathrm{sen}}$ and $\lambda_{\mathrm{cap}}$ are used to balance losses of two levels. This design encourages the prompt to incrementally learn richer semantics, enhancing the performance of prompt tuning. The overall loss $\mathcal{L}$ combines autoregressive cross-entropy loss $\mathcal{L}_{\mathrm{autoregressive}}$ and semantic alignment loss $\mathcal{L}_{\mathrm{semantic}}$:

$$\mathcal{L} = \mathcal{L}_{\mathrm{autoregressive}} + \mathcal{L}_{\mathrm{semantic}}. \tag{9}$$

## 4 EXPERIMENT

### 4.1 EXPERIMENT SETTINGS

**Datasets** Based on ImageNet (Krizhevsky et al., 2017), we construct a multi-level tuning text dataset to support multi-level semantic refinement. Sentence-level text: a fixed template, *"a photo of {class_name}"*, is used to provide relatively comprehensive semantics across the 1000 categories. Caption-level text: detailed captions generated by BLIP-2 (Li et al., 2023a) provide fine-grained visual attributes and semantic information. Furthermore, to assess the transferability and generality of MVP, we conduct additional experiments on Food101 (Bossard et al., 2014), RESISC45 (Cheng et al., 2017), SUN397 (Sun et al., 2023), and MS-COCO (Lin et al., 2014).

**Implementation Details** We implement MVP on VAR with 16, 20, 24, 30, 36 layers and follow the experimental settings of VAR. For ablation and analysis, we also transfer MVP to other VAR-like models such as HART (Tang et al., 2024) and Infinity (Han et al., 2025). The AdamW (Loshchilov & Hutter, 2017) optimizer is employed for training. Notably, for class-to-image generation, we discard the first-scale prompt to prevent interference with class embeddings. All evaluations are conducted on a single NVIDIA A100 GPU with 80 GB of memory. Appendix D includes more details.

### 4.2 MAIN RESULTS

**Improve VAR's Class-to-Image Generation Quality** We evaluate MVP on VAR with depths of 16, 20, 24, and 30 to generate $256 \times 256$ images on ImageNet. Table 1 presents a comprehensive comparison between MVP, VAR, and other types of generation models. As observed, compared to VAR, MVP introduces only minimal parameters while achieving improvements in FID and IS, even with some showing marked improvements. For example, MVP reduces FID by 5% compared to VAR-d30. In comparison with other types of generation models, MVP also maintains the advantages

of VAR while further extending its lead, achieving notable performance gains while training only less than 1% of the parameters. Moreover, we also employ MVP at the depth of 36 for image generation at a higher resolution of $512 \times 512$ on ImageNet. As shown in Table 2, MVP also surpasses VAR as well as other types of generation models.

Table 1: Comparisons on class-to-image generation on ImageNet. Evaluation metrics include Fréchet Inception Distance (FID), Inception Score (IS) and inference time (s). Precision and recall jointly assess the fidelity–diversity trade-off of generated images. The suffix '-re' denotes rejection sampling. '↓' and '↑' indicate that lower or higher values are preferable.

| Type | Model | Param | FID↓ | IS↑ | Precision↑ | Recall↑ | Time |
|---|---|---|---|---|---|---|---|
| GAN | BigGAN Brock et al. (2018) | 112M | 6.95 | 224.5 | 0.89 | 0.38 | – |
| GAN | GigaGAN Kang et al. (2023) | 569M | 3.45 | 225.5 | 0.84 | 0.61 | - |
| GAN | StyleGAN-XL Sauer et al. (2022) | 166M | 2.30 | 265.1 | 0.78 | 0.53 | 0.3 |
| Diffusion | ADM Dhariwal & Nichol (2021) | 554M | 10.94 | 101.0 | 0.69 | 0.63 | 168 |
| Diffusion | CDM Ho et al. (2022) | - | 4.88 | 158.7 | - | - | - |
| Diffusion | LDM-4 Rombach et al. (2022) | 400M | 3.60 | 247.7 | - | - | - |
| Diffusion | DiT-XL/2 Peebles & Xie (2023) | 675M | 2.27 | 278.2 | 0.83 | 0.57 | 31 |
| *Masked* AR | MaskGIT Chang et al. (2022) | 227M | 6.18 | 182.1 | 0.80 | 0.51 | 0.5 |
| *Masked* AR | MaskGIT-re Li et al. (2023b) | 227M | 4.02 | 355.6 | - | - | |
| *Masked* AR | MAGE Li et al. (2024) | 230M | 6.93 | 195.8 | - | - | - |
| *Next-token* AR | VQGAN Esser et al. (2021b) | 227M | 18.65 | 80.4 | 0.78 | 0.26 | 19 |
| *Next-token* AR | VQGAN-re Yu et al. (2021) | 1.4B | 5.20 | 280.3 | - | - | 24 |
| *Next-token* AR | VQGAN (1.4B) Esser et al. (2021b) | 1.4B | 15.76 | 74.3 | - | - | 25 |
| *Next-token* AR | ViT-VQGAN Yu et al. (2021) | 1.7B | 4.17 | 175.1 | - | - | > 24 |
| *Next-token* AR | ViT-VQGAN-re Yu et al. (2021) | 1.7B | 3.04 | 227.4 | - | - | > 24 |
| *Next-token* AR | RQTran Lee et al. (2022) | 3.8B | 7.55 | 80.4 | 0.78 | 0.26 | 21 |
| *Next-token* AR | RQTran-re Lee et al. (2022) | 3.8B | 3.80 | 323.7 | - | - | 21 |
| *Next-token* AR | LlamaGen-B Sun et al. (2024) | 111M | 5.46 | 193.6 | 0.83 | 0.45 | - |
| *Next-token* AR | LlamaGen-L Sun et al. (2024) | 343M | 3.81 | 248.3 | 0.83 | 0.52 | - |
| *Next-token* AR | LlamaGen-XL Sun et al. (2024) | 775M | 3.39 | 227.1 | 0.81 | 0.54 | - |
| *Next-token* AR | LlamaGen-XXL Sun et al. (2024) | 1.4B | 3.09 | 253.6 | 0.83 | 0.53 | - |
| *Next-scale* AR | VAR-*d16* Tian et al. (2024) | 310M | 3.61 | 225.6 | 0.81 | 0.52 | 0.4 |
| *Next-scale* AR | VAR-*d20* Tian et al. (2024) | 600M | 2.67 | 254.4 | 0.81 | 0.57 | 0.5 |
| *Next-scale* AR | VAR-*d24* Tian et al. (2024) | 1.0B | 2.17 | 271.9 | 0.81 | 0.59 | 0.6 |
| *Next-scale* AR | VAR-*d30* Tian et al. (2024) | 2.0B | 2.14 | 275.4 | 0.80 | 0.60 | 1 |
| *Next-scale* AR | MVP-*d16* | 310.4M | 3.46 | 247.4 | 0.83 | 0.52 | 0.4 |
| *Next-scale* AR | MVP-*d20* | 601M | 2.63 | 276.5 | 0.82 | 0.55 | 0.6 |
| *Next-scale* AR | MVP-*d24* | 1.02B | 2.13 | 292.9 | 0.81 | 0.58 | 0.6 |
| *Next-scale* AR | MVP-*d30* | 2.01B | 2.03 | 289.4 | 0.81 | 0.59 | 1 |

Table 2: Comparisons on class-to-image generation with $512 \times 512$ resolution on ImageNet .

| Type | Model | FID↓ | IS↑ | Time |
|---|---|---|---|---|
| GAN | BigGAN Brock et al. (2018) | 8.43 | 177.9 | – |
| Diffusion | ADM Dhariwal & Nichol (2021) | 23.24 | 101.0 | – |
| Diffusion | DiT-XL/2 Peebles & Xie (2023) | 3.04 | 240.8 | 81 |
| Masked autoregressive | MaskGIT Chang et al. (2022) | 7.32 | 156.0 | 0.5 |
| *Next-token* autoregressive | VQGAN Esser et al. (2021b) | 26.52 | 66.8 | 25 |
| *Next-scale* autoregressive | VAR-*d36* Tian et al. (2024) | 2.63 | 303.2 | 1 |
| *Next-scale* autoregressive | MVP-*d36* | 2.47 | 317.4 | 1 |

**Expand Text-to-Image Generation Capability** Since MVP extends the text-to-image generation capability of VAR, we compare MVP with VAR-CLIP (Zhang et al., 2024a), a fully pretrained method targeting the same extension task. In addition, to explore the superiority of the perturbation-based MVP, we also compare it with LoRA, an adapter-based parameter-efficient fine-tuning method. As shown in Tab. 3, MVP achieves an excellent trade-off between generation quality and training efficiency. Specifically, MVP requires only 0.46% of the training parameters and 0.54% of the training time of VAR-CLIP, while maintaining a competitive FID and surpassing VAR-CLIP

in CLIP-Score. Meanwhile, compared with LoRA, MVP also achieves overall superiority in both efficiency and performance. This highlights that MVP is a simple yet efficient prompt tuning method.

Table 3: Comparison of different tuning methods. TP is the number of trainable parameters.

| Method | TP | FID↓ | CLIP-Score↑ | GPU-Hours↓ |
|--------|------|-------|-------------|------------|
| VAR-CLIP | 310M | 11.26 | 28.55 | 4782 |
| VAR+LoRa | 4.6M | 14.8 | 29.23 | 58 |
| VAR+MVP | 1.45M | 13.50 | 30.48 | 26 |

Table 4: Comparison of multi-scale and prefilled prompt. Memory (GB) is peak GPU memory.

| Method | FID↓ | IS↑ | Memory↓ |
|--------|------|-------|---------|
| VAR-*d16* | 3.61 | 225.6 | – |
| + Prefilled Prompt | 3.51 | 238.4 | 23.2 |
| + MVP (Ours) | 3.46 | 247.4 | 14.5 |

## 4.3 ANALYSIS & ABLATIONS

**Multi-scale Prompt vs. Prefilled Prompt** We evaluate MVP against the traditional prefilled prompt on class-to-image generation to compare the advantages between embedding-based and perturbation-based prompt tuning. All experiments are conducted with the 16-depth VAR backbone, with the scale threshold $\tau$ fixed at 20 and the same number of visual prompt tokens. As shown in Table 4, while both methods improve the generation quality, MVP achieves better FID and IS than the prefilled prompt. Moreover, our fine-tuning process incurs lower memory overhead, as our strategy avoids increasing the token sequence length.

**Effect of Prompt Position** To validate the effectiveness of our prompt position design in MVP, which injects prompts at outermost square frame of feature maps, we compare it with four alternative position designs with an identical prompt token budget: (*i*) Random: Prompt tokens are randomly placed across the feature map; (*ii*) Innermost: Prompt tokens are first placed at the center of the feature map (the innermost square frame) and then expand outward until the prompt token budget is reached; (*iii*) Center-to-Outer: Prompt tokens are placed starting from the innermost frame, then skipping one frame before placing the next frame of prompt tokens, and so on, forming concentric prompt frames across the feature map. (*iv*) Sub-Outermost: Prompt tokens are placed from the second outermost square frame and extend outward if the prompt budget is not yet filled. All alternative position designs maintain an equal prompt token budget and are tested on VAR backbones with different depths (d16 and d30) for class-conditional generation. For fair comparison and to reduce randomness effects, the Random Placement variant is repeated three times with different random seeds, and the averaged results are reported. As summarized in Table 5, the outermost square frame strategy consistently outperforms other strategies, indicating that our prompt position design minimizes distributional distortion of the pretrained models while providing effective semantic guidance.

Table 5: Performance across different prompt position designs on class-conditional generation. '*' indicates mean results over 3 runs with different random seeds.

| Depth | Prompt Position | FID↓ | IS↑ |
|-------|-----------------|-------|--------|
| 16 | Ours | **3.46** | **247.4** |
| 16 | Random | 3.69* | 238.2* |
| 16 | Innermost | 3.68 | 240.0 |
| 16 | Center-to-Outer | 3.64 | 237.8 |
| 16 | Sub-Outermost | 3.51 | 243.1 |
| 30 | Ours | **2.03** | **289.4** |
| 30 | Random | 2.16* | 281.7* |
| 30 | Innermost | 2.13 | 283.3 |
| 30 | Center-to-Outer | 2.14 | 279.6 |
| 30 | Sub-Outermost | 2.08 | 285.2 |

**Effect of First-scale Prompt** We examine the impact of introducing the visual prompt at the first scale. As shown in Table 6, introducing the prompt at the first scale significantly degrades class-to-image performance, especially in terms of IS, indicating disruption to the VAR backbone's learned class embeddings. In contrast, first-scale prompting improves text-to-image generation, facilitating alignment between text and class embeddings.

Table 6: Ablation of prompts at the 1st scale.

| Depth | Injection | FID↓ | IS↑ |
|-------|-----------|-------|--------|
| 16 | ✓ | 3.57 | 221.3 |
| 16 | ✗ | **3.46** | **247.4** |
| 20 | ✓ | 2.66 | 237.6 |
| 20 | ✗ | **2.63** | **276.5** |
| 24 | ✓ | 2.15 | 275.4 |
| 24 | ✗ | **2.13** | **292.9** |

**Effect of Threshold** $\tau$ To determine a suitable scale threshold for MVP, we conduct ablation studies on VAR backbones with depth 16, 20, 24 and 30. As shown in Tab. 7, appropriate thresholds can achieve a good balance: they provide sufficient prompt capacity to improve FID and IS, while avoiding the redundancy and instability that occur with too small or overly large thresholds.

Table 7: Ablation study on prompt scale threshold $\tau$ across VAR backbones with different depths.

| Depth | Threshold $\tau$ | FID↓ | IS↑ | Depth | Threshold $\tau$ | FID↓ | IS↑ |
|-------|------------------|------|-----|-------|------------------|------|-----|
| 16 | 4 | 3.58 | 231.1 | 24 | 4 | 2.19 | 275.8 |
| 16 | 12 | 3.49 | 238.3 | 24 | 12 | 2.15 | 281.4 |
| 16 | 20 | **3.46** | **247.4** | 24 | 20 | **2.13** | **292.9** |
| 16 | 28 | 3.47 | 248.8 | 24 | 28 | 2.13 | 291.4 |
| 20 | 4 | 2.71 | 262.4 | 30 | 12 | 2.15 | 284.1 |
| 20 | 12 | 2.66 | 271.7 | 30 | 20 | 2.12 | 281.6 |
| 20 | 20 | **2.63** | **276.5** | 30 | 28 | **2.03** | **292.9** |
| 20 | 28 | 2.62 | 271.3 | 30 | 36 | 2.05 | 297.3 |

Table 8: Comparison of different PEFT methods for cross-dataset transfer on metrics FID↓.

| Method | Trainable Params (%) | Mean | SUN397 | Food101 | Resisc |
|--------|----------------------|------|--------|---------|--------|
| VAR (fine-tuning) | 100 | 37.27 | 23.85 | 30.07 | 57.90 |
| VAR (LoRA) | 0.26 | 30.87 | 22.73 | 29.65 | 40.22 |
| VAR (QLoRA) | 0.26 | 30.91 | 22.67 | 29.65 | 40.41 |
| VAR (MVP) | 0.17 | **29.72** | **22.42** | **28.76** | **37.97** |

**Comparison with Other PEFT Methods**    To evaluate the transferability and generality of MVP across different data distributions, we compare MVP with LoRA and QLoRA on additional datasets: SUN397, RESISC45 and Food101. All experiments are conducted on VAR-d24 with a single epoch of fine-tuning. As shown in Table 8, MVP achieves superior performance to other PEFT methods such as LoRA and QLoRA, while using fewer trainable parameters. Notably, MVP demonstrates a clear advantage under significant domain shifts, such as on the RESISC dataset. These results highlight MVP as an efficient and robust parameter-efficient tuning strategy.

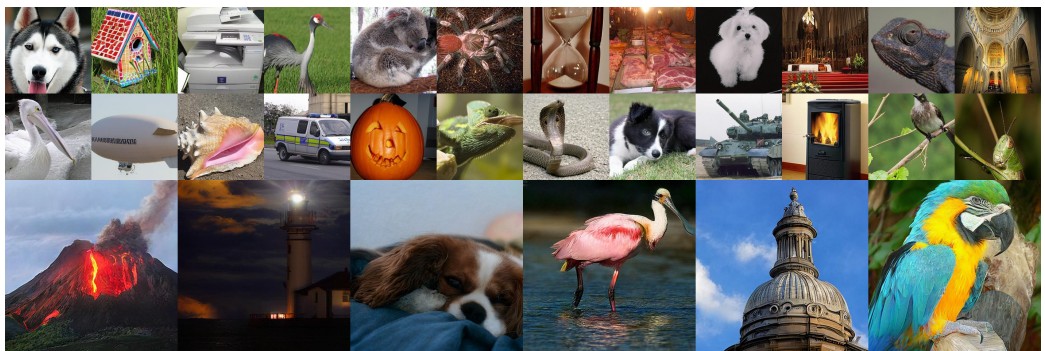

Figure 2: Visualization of class-to-image samples generated use MVP. The first two rows show results at 256×256 resolution, and the third row shows results at 512×512 resolution.

## 4.4 VISUALIZATION

**Class-to-Image Generation**    In Figure 2, randomly selected samples on ImageNet show that MVP generates images with high visual fidelity and diversity. More visualization is provided in Appendix.

**Text-to-Image Generation**    As shown in Figure 3, visualization of text-to-image generation (24 depths, 26 GPU-Hours, a single 80G A100) demonstrates that MVP effectively enhances text–image alignment, expanding VAR's text-to-image generation capability. Notably, both MVP and VAR-CLIP are built upon the same VAR backbones: while VAR-CLIP (depth=16) requires full pretraining with 4782 GPU-Hours on 48 A100 80GB GPUs, MVP attains competitive performance with less than 1% of its computational cost. Moreover, this efficiency advantage is consistently preserved even when applied to a deeper backbone, highlighting the practicality of our fine-tuning strategy.

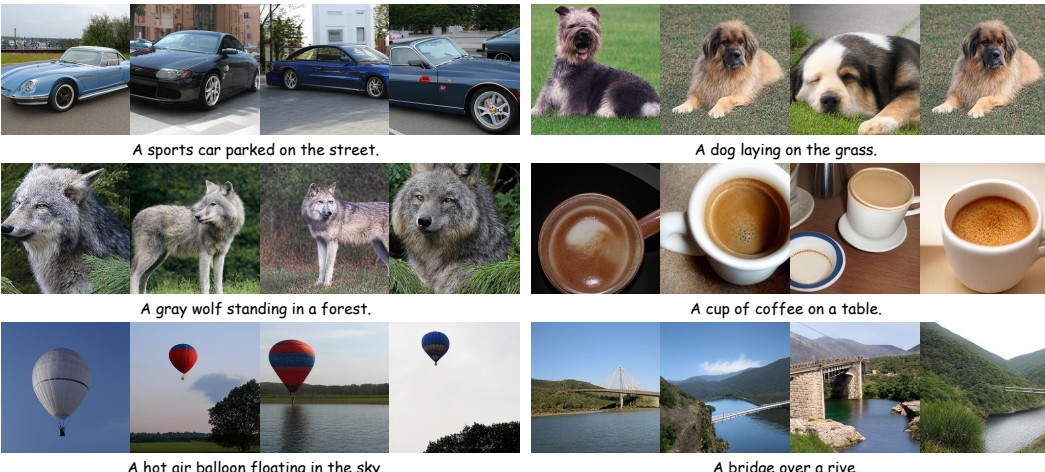

Figure 3: Visualization of text-to-image generation samples at 256×256 resolution using MVP.

## 5 CONCLUSION

In this paper, we propose MVP, a multi-scale visual prompt method with planar concept and efficient information propagation tailored to VAR. By introducing prompt in the outermost square frame and increasingly detailed tuning text, MVP enables effective prompt learning of rich semantics and task features at a relatively low computational cost. Moreover, MVP not only significantly improves performance on the class-to-image generation, but also extends VAR's text-to-image generation capability. This offers a novel and promising direction for visual autoregressive generation.

### ETHICS STATEMENT

This study is conducted exclusively on publicly available benchmark datasets (ImageNet, Food101, RESISC45, SUN397, and MS-COCO), which are widely adopted in the computer vision research community. These datasets contain no personally identifiable information or sensitive data. The proposed methods focus on achieving class-conditional and text-to-image generation within these benchmarks through perturbation-based prompt design. However, we do not foresee direct negative societal impacts, but acknowledge that generative models may be misused for producing misleading or harmful content. We encourage responsible usage of our models and provide detailed descriptions of implementation and training settings in the appendix to support reproducibility and transparency. This research adheres to the ICLR Code of Ethics.

### REPRODUCIBILITY STATEMENT

We have taken extensive measures to ensure the reproducibility of our results. The model architecture and implementation details are provided in Section 4.1 of the main text, while comprehensive training configurations and hyperparameter settings are described in Appendix D. All datasets used in our experiments are publicly available. In addition, we provide an anonymous supplementary link to our source code, which includes the full training and inference scripts, to further facilitate independent verification of our findings.

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
