APPENDIX

## A    THE USE OF LARGE LANGUAGE MODELS

In preparing this research, we made limited use of large language models (LLMs) as assistive tools. Specifically, LLMs were employed for:

1. **Writing assistance**: LLMs were used to improve grammar, enhance clarity, and ensure an appropriate academic tone in certain parts of the manuscript.
2. **Editing and formatting**: To support professional presentation, LLMs were used to generate LaTeX table/figure templates and to refine the structure of paragraphs.

It should be emphasized that LLMs were not involved in the conception of core research ideas, the design or execution of experiments, the analysis of results, or the formulation of scientific conclusions. All technical contributions, algorithms, experiments, and analyses are original and were entirely carried out by the authors.

The authors take full responsibility for the content of this paper. LLMs were not considered contributors or co-authors, and their usage was strictly limited to the auxiliary functions described above.

## B    MORE MAIN RESULTS

Beyond the main experiments presented above, we further provide additional evaluations to demonstrate the robustness and generality of MVP.

**Evaluation on other VAR-based T2I models.**    We further evaluate the transferability of MVP by applying it to VAR-based text-to-image models, namely Infinity and HART, both of which are pretrained on high-quality aesthetic datasets. To evaluation robustness under distribution shifts, we fine-tune these models on MS-COCO, a dataset featuring more realistic and diverse scenes that differ substantially from their original training distribution. As shown in Tab. 9, directly performing full fine-tuning on such distribution-shifted data often induces catastrophic forgetting, resulting in notable degradation in both FID and CLIP-Score. In contrast, MVP achieves consistent improvements while requiring only a fraction of the parameters to be updated, effectively preserving the pretrained generative priors and adapting to the new domain with minimal overhead. These results highlight the practicality of MVP as a lightweight and robust transfer approach for VAR-based T2I models.

Table 9: Transferability of MVP to other VAR-based text-to-image models (HART Tang et al. (2024) and Infinity Han et al. (2025)) under distribution shifts from aesthetic datasets to MS-COCO.

| Method | Trainable Params (%) | FID ↓ | CLIP-score↑ |
|---|---|---|---|
| HART-0.7B | - | 36.2 | 0.22 |
| HART-0.7B (Full FT) | 100 | 56.4 | 0.17 |
| HART-0.7B (MVP) | 0.27 | 31.3 | 0.24 |
| Infinity-2B | - | 36.9 | 0.23 |
| Infinity-2B (Full FT) | 100 | 63.2 | 0.20 |
| Infinity-2B (MVP) | 0.13 | 29.9 | 0.24 |

## C    MORE ANALYSIS & ABLATIONS

In this section, we provide additional ablation studies to further analyze the effectiveness of each component in our method.

### C.1    POTENTIAL CONCERN OF DISTRIBUTION DISTORTION.

One potential concern is that perturbation-based prompts in MVP may distort the pretrained VAR backbone's distribution, thereby impairing generation quality and casting doubt on the viability of

perturbation-driven prompting. To better understand this effect, we performed ablation studies from two complementary perspectives.

**Impact of the Gating Mechanism.** Gating is commonly expected to mitigate distribution distortion by constraining the magnitude of perturbation-based prompts. However, our experiments on improve class-conditional generation reveal the opposite: while gating reduces prompt strength, it also limits the method's ability to provide effective guidance and can even degrade the fidelity and diversity of generated results (Tab. 10). This behavior likely stems from MVP's multi-scale propagation of semantic and structural signals. When the perturbation strength is overly constrained, the prompts degenerate into weak noise that fails to deliver clear guidance and instead interferes with the pretrained features of the backbone. Consequently, although the gating mechanism is intended to reduce distributional shift, it paradoxically disrupts the learned representations and diminishes the overall effectiveness of our method.

Table 10: Comparison between the default MVP design and its gated variant on enhancing the generative capability of VAR at different depths.

| Model | Depth | FID↓ | IS↑ | Precision↑ | Recall↓ |
|---|---|---|---|---|---|
| VAR | 24 | 2.17 | 271.9 | 0.81 | 0.59 |
| MVP | 24 | 2.13 | 292.9 | 0.81 | 0.58 |
| MVP (Gating) | 24 | 2.17 | 273.9 | 0.80 | 0.59 |
| VAR | 30 | 2.14 | 275.4 | 0.80 | 0.60 |
| MVP | 30 | 2.03 | 289.4 | 0.81 | 0.59 |
| MVP (Gating) | 30 | 2.25 | 274.1 | 0.80 | 0.59 |

## C.2    ABLATION ON INJECTION INTO HIDDEN STATES

A potential concern is the prompts injection applied only at the input stage might gradually diminish during propagation through the transformer layers, thereby weakening its effect. To mitigate this, we assess the impact of injecting prompts into hidden states during autoregressive modeling across varying model depths and injection frequencies. As shown in Tab. 11, this strategy leads to performance degradation rather than improvement. Specifically, increasing the injection frequency consistently causes both FID and IS to deteriorate across different backbone depths. This result further demonstrates that our method achieves optimal effectiveness with its minimalist design.

Table 11: Impact of injecting projector-processed visual prompt tokens at varying frequencies into intermediate layers of the transformer.

| Depth | Injection | Freq. | FID↓ | IS↑ | Depth | Injection | Freq. | FID↓ | IS↑ |
|---|---|---|---|---|---|---|---|---|---|
| 16 | ✗ | - | **3.46** | **247.4** | 24 | ✗ | - | **2.13** | **292.9** |
| 16 | ✓ | 1 | 3.51 | 242.9 | 24 | ✓ | 1 | 2.17 | 286.1 |
| 16 | ✓ | 2 | 3.69 | 217.4 | 24 | ✓ | 2 | 2.20 | 274.5 |
| 16 | ✓ | 3 | 3.77 | 209.4 | 24 | ✓ | 3 | 2.28 | 263.7 |
| 20 | ✗ | - | **2.63** | **276.5** | 30 | ✗ | - | **2.03** | **289.4** |
| 20 | ✓ | 1 | 2.70 | 264.7 | 30 | ✓ | 1 | 2.11 | 291.8 |
| 20 | ✓ | 2 | 2.81 | 254.1 | 30 | ✓ | 2 | 2.13 | 295.3 |
| 20 | ✓ | 3 | 2.87 | 231.3 | 30 | ✓ | 3 | 2.26 | 281.4 |

## C.3    ABLATION ON PROMPT MODULATION

A natural concern is that adopting a single, fixed prompt shared across all classes may hinder the model's ability to capture class-specific nuances, raising the question of whether adaptive prompts would better accommodate diverse generation tasks. To examine this, we designed two modulated variants: (i) a FiLM-based modulation that predicts feature-wise scale and shift parameters from the class condition, and (ii) a lightweight cross-attention module that injects class information into the original prompt sequence.

The experimental results are summarized in Tab. 12. While these variants introduce additional flexibility, they also substantially increase the number of trainable parameters and the difficulty of optimization. Under the same training budget (e.g., 1 epoch), both variants underperform compared to our default design, showing weaker FID and IS scores. Extending training to more epochs can indeed improve the generation quality of these variants, but such longer schedules run counter to the efficiency principle of prompt tuning, whose key advantage lies in rapid adaptation with minimal computational cost.

Table 12: Impact of injecting projector-processed visual prompt tokens at varying frequencies into intermediate layers of the transformer.

| Depth | FiLM | FID↓ | IS↑ | Epochs | | Depth | CA. | FID↓ | IS↑ | Epochs |
|---|---|---|---|---|---|---|---|---|---|---|
| 16 | ✗ | 3.46 | 247.4 | 1 | | 16 | ✗ | 3.46 | 247.4 | 1 |
| 16 | ✓ | 3.74 | 228.2 | 1 | | 16 | ✓ | 3.97 | 231.6 | 1 |
| 16 | ✓ | 3.61 | 239.7 | 3 | | 16 | ✓ | 3.67 | 242.3 | 3 |
| 20 | ✗ | 2.63 | 276.5 | 1 | | 20 | ✗ | 2.63 | 276.5 | 1 |
| 20 | ✓ | 2.74 | 259.1 | 1 | | 20 | ✓ | 2.88 | 248.3 | 1 |
| 20 | ✓ | 2.68 | 267.7 | 3 | | 20 | ✓ | 2.77 | 271.2 | 3 |

# D   MORE DETAIL

## D.1   MORE IMPLEMENTATION DETAIL

In the VAR backbone configurations, the detailed training settings, including learning rate, batch size, number of epochs, and other hyperparameters, are provided in Tab. 13 for the class-to-image task and in Tab. 14 for the text-to-image task. Notably, for the experiments on improving class-to-image generation, we discard the first-scale prompts to prevent interference with the pretrained class embeddings and maintain the conditioning integrity.

Table 13: Implementation detail of MVP for class-to-image

| backbone | VAR-d16 | VAR-d20 | VAR-d24 | VAR-d30 | VAR-d36 |
|---|---|---|---|---|---|
| $\tau$ | 20 | 20 | 20 | 28 | 28 |
| first-scale prompt | ✗ | ✗ | ✗ | ✗ | ✗ |
| optimizer | | | AdamW | | |
| AdamW ($\beta_1$, $\beta_2$) | | | (0.9, 0.95) | | |
| learning rate | 1e-3 | 1e-3 | 1e-3 | 5e-4 | 8e-5 |
| weight decay | 5e-2 | 5e-2 | 1e-2 | 1e-2 | 1e-2 |
| batch size | 132 | 124 | 82 | 58 | 12 |
| epoch | 1 | 1 | 1 | 1 | 1 |

Table 14: Implementation detail of MVP for text-to-image

| backbone | VAR-d16 | VAR-d20 | VAR-d24 | VAR-d30 | HART-0.7B | Infinity-2B |
|---|---|---|---|---|---|---|
| $\tau$ | 20 | 28 | 28 | 36 | 36 | 44 |
| first-scale prompt | ✓ | ✓ | ✓ | ✓ | ✗ | ✗ |
| optimizer | | | AdamW | | | |
| AdamW ($\beta_1$, $\beta_2$) | | | (0.9, 0.99) | | | |
| learning rate | 1e-4 | 1e-4 | 7e-5 | 8e-5 | 1e-4 | 1e-4 |
| weight decay | 5e-2 | 5e-2 | 1e-2 | 1e-2 | 5e-2 | 5e-2 |
| batch size | 148 | 124 | 82 | 58 | 12 | 8 |
| epoch | 1 | 1 | 1 | 1 | 1 | 1 |

### D.2 TRAINING LOSS DETAILS

**Training loss details of MVP and VAR**  To further complement the results presented in Table 8, we analyze the training loss dynamics of MVP and full-parameter VAR fine-tuning across three datasets (SUN397, Food101, and RESISC). As illustrated in Fig. 4, MVP consistently converges faster than full-parameter VAR, achieving both a steeper early-stage loss drop and a lower final loss. For clarity, we visualize both the raw loss and an exponentially smoothed version (smoothing factor value is 0.65). The results across all datasets show substantially more stable optimization behavior and a more favorable convergence trajectory with MVP. These observations further validate MVP's strong transferability and robust adaptation ability compared with standard full-parameter VAR fine-tuning approaches.

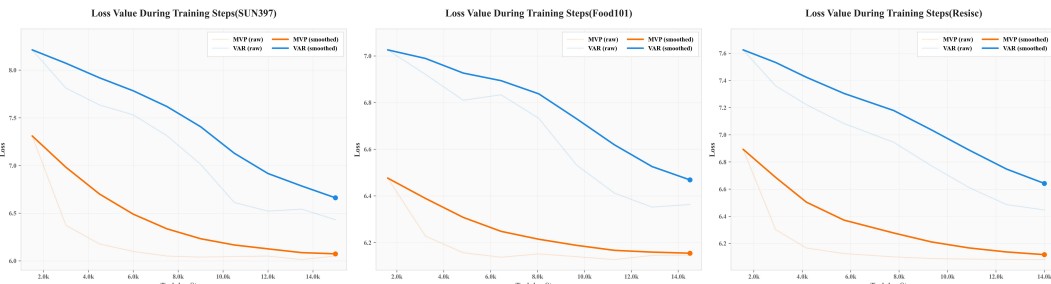

Figure 4: Training loss details for MVP vs. full-parameter VAR fine-tuning.

**Training loss details for different prompt placement strategies**  To more precisely examine the differences in efficiency and performance across different prompt position designs, we visualize the training loss behavior of our design in MVP and four alternative position designs under an identical prompt token budget. As shown in Fig. 5, the resulting gaps $\Delta$ clearly show that our outermost-frame position design not only achieves the fastest convergence but also yields the lowest final loss across all variants. These results provide strong optimization-level evidence that our prompt placement design imposes minimal perturbation on pretrained representations while offering the most effective semantic conditioning.

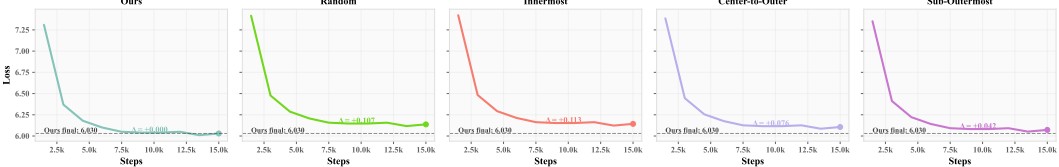

Figure 5: Training loss details for different prompt position designs.

### D.3 MORE THEORETICAL JUSTIFICATION DETAILS

To complement the simplified analysis in Section 3.1, we provide a more complete theoretical rationale explaining why the outermost-frame region constitutes the most informative and stable location for placing prompt tokens in multi-scale VAR generation.

**Propagation Distance and Signal Attenuation Model**  We begin by formalizing the distance-based attenuation model of prompt influence. Consider the feature map as a 2D grid (or graph) of tokens $V = (x, y) \mid 1 \leq x, y \leq S_t$ at a given scale $t$. Let $d((x, y), (u, v))$ denote the distance between two token positions. For theoretical generality, $d$ can be any appropriate metric on the grid (e.g. Manhattan distance, Euclidean, or Chebyshev distance), but we will often treat it as the shortest-path distance on the grid graph (equivalent to Manhattan distance if only orthogonal moves are allowed). We suppose that a prompt token at position $p$ imparts a perturbation signal $\delta$ that propagates to other tokens with a strength that decays as a function of distance. Formally, let $f(d)$

be a monotonically decreasing attenuation function (with $f(0) = 1$ and $0 < f(d) < 1$ for $d > 0$). A simple choice in our initial analysis was an exponential decay $f(d) = \alpha^d$ for some $0 < \alpha < 1$, but more generally one could consider, for example, an exponential $f(d) = e^{-\lambda d}$, a power-law $f(d) = (1 + \beta d)^{-1}$, or other diffusion-based kernels. The key property is that influence diminishes with propagation distance.

If a single prompt token is introduced at some source location $u \in V$ with perturbation magnitude $\delta$, the impact it has on a target token $v \in V$ can be modeled as $I_{u \to v} = \delta, f!\big(d(u, v)\big)$. When multiple prompt tokens are present, their effects can be assumed to superpose (e.g. additively, if we treat small perturbations linearly). Thus, for a set of prompt positions $B \subset V$, the total influence on $v$ is:

$$I_B(v) = \sum_{u \in B} \delta_u \, f(d(u, v)),$$

where $\delta_u$ is the perturbation added at prompt $u$ (for simplicity one may take all $\delta_u = \delta$). In general, $I_B(v)$ will be stronger for targets $v$ that are closer to some prompt token and weaker for those far from all prompts. We can thus formalize two objectives for prompt placement:

(1).Minimal Central Impact: Avoid placing prompts too close to the critical center region of the image, so that any central token is at a large distance from all prompts. This keeps $f(d)$ small for center distances, mitigating disruptive changes to the core visual content. (2)Maximal Propagation Coverage: Ensure that every token in the feature map lies reasonably close to at least one prompt, so that prompt signals can efficiently reach all parts of the map. In other words, we want to minimize the worst-case (or average) distance from any token to the nearest prompt, thereby maximizing the minimum influence $I_B(v)$ across the map.

These goals are in tension: keeping prompts far from the center suggests placing them at the periphery, but doing so might increase distances to some other tokens. We next analyze this trade-off formally and show that an outermost-frame placement of prompts achieves an optimal balance under these criteria.

**Boundary vs. Interior Prompt Placement: Distance Analysis**   Consider a decomposition of the $S_t \times S_t$ feature map into concentric square frames (or layers) around the center. Define Frame 0 as the set of all tokens on the outer boundary of the map (positions where $x = 1$, $x = S_t$, $y = 1$, or $y = S_t$). Frame 1 is the next inward layer (the "sub-outermost" border), and so on, until Frame $N$ which contains the central token(s). By construction, $N = \lfloor \frac{S_t - 1}{2} \rfloor$, i.e. there are $N$ concentric frames inward from the boundary to the center. Each frame index $n$ can be thought of as the Chebyshev distance (infinite norm distance) of those tokens from the image center. Now suppose we introduce prompt tokens on Frame $c$ (meaning all prompt tokens lie in that concentric layer at distance $c$ from the boundary). What is the distance from these prompts to the center, and to other tokens? Two key observations can be made:

- Distance to Center: All prompt tokens on Frame $c$ are a distance of $(N - c)$ away from the center frame (in terms of frame index difference). In fact, the minimum distance from the center to any prompt in Frame $c$ is $N - c$ (achieved along a straight line from the center to the prompt layer). Thus, a prompt at frame $c$ induces a central impact of roughly $I_{center} \approx \delta, f(N - c)$. If we use the exponential model $f(d) = \alpha^d$, this recovers the earlier result that a prompt in frame $n$ has impact $\delta, \alpha^{N-n}$ on the center, which increases rapidly as $n$ grows closer to $N$ (i.e. as the prompt moves inward). Keeping prompts in the outermost frame ($c = 0$) maximizes the center distance $N - 0 = N$, yielding minimal impact on the central tokens $I_{center} \approx \delta, \alpha^N$. By contrast, any non-boundary placement ($c > 0$) would put prompt tokens closer to the center (distance $N - c$ with $N - c < N$), leading to significantly larger direct impacts on central features (e.g. a frame $c$ prompt gives $\delta, \alpha^{N-c}$, and since $N - c < N$, we get $\alpha^{N-c} \gg \alpha^N$ for $\alpha \in (0, 1)$). In the extreme case of a prompt at the center itself ($c = N$), the distance to center is 0 and the central impact is maximized (no attenuation, $f(0) = 1$) – this would heavily "corrupt" the core visual features, which is exactly what we must avoid. Therefore, placing prompts on the outer boundary is theoretically optimal for protecting the image center: it maximizes the minimum distance from any prompt to the central region, minimizing unwanted perturbation of semantically critical center content.

- Distance to Other Tokens (Coverage): We must also ensure that prompt signals can reach and influence all other tokens in the feature map efficiently. For a given token $v = (x, y)$ in the map, let $\text{dist}B(v) = \min u \in B d(u, v)$ denote the distance from $v$ to the nearest prompt in the set $B$. We seek to make $\text{dist}_B(v)$ as small as possible for all $v$. If $B$ is the outermost frame ($B = B_t$ in scale $t$, using the notation $B_t$ for the set of all boundary positions), then every non-prompt token lies somewhere inside the boundary. Intuitively, any interior location will be adjacent (in some direction) to the boundary after a certain number of steps outward. In fact, for an arbitrary token at $(x, y)$, the minimal Manhattan distance to the outer frame is given by:

$$\text{dist}_{B_t}(x, y) = \min\{\, x - 1,\ S_t - x,\ y - 1,\ S_t - y \,\}. \tag{10}$$

  This formula says that the distance to the boundary is determined by whichever edge (top, bottom, left, or right) is nearest to $(x, y)$. For example, if a token is 5 cells from the top edge, 10 from the bottom, 3 from the left edge, and 8 from the right edge, then its nearest prompt on the outer frame is 3 away (via the left border). The worst-case distance from any token to the outer-frame prompts is achieved at the geometric center of the map: a central token is farthest from all edges. Indeed, from Eq. 10 one can show the maximum distance is

$$D_{\max}(B_t) = \max_{(x,y)\in V} \text{dist}_{B_t}(x, y) = \left\lfloor \frac{S_t - 1}{2} \right\rfloor = N,$$

  attained at the center ($x = \lceil S_t/2 \rceil,\,; y = \lceil S_t/2 \rceil$). Thus, with outermost-frame prompts, no token is more than $N$ steps away from some prompt. In terms of the attenuation function, the minimum prompt influence on any token $v$ is at least $\delta, f(D_{\max}) = \delta, f(N)$. Using the exponential model as an example, the weakest-controlled token (at the center) still receives a small but nonzero signal $\delta, \alpha^N$. All other tokens are closer than $N$ steps to the boundary and hence receive stronger prompt influence than the center does. In other words, placing prompts on the boundary yields full coverage of the map with a radius $N$: prompt signals need at most $N$ steps to reach any location.

It is insightful to contrast this with alternative placements. If prompts were placed at the center instead (Frame $N$), the situation essentially inverts: the center prompt directly affects itself (distance 0), but tokens on the outer border are now farthest (roughly $N$ steps away) and would experience the weakest influence. In fact, a single central prompt also has a worst-case distance of $N$ – the corners/edges of the map are $N$ units away from the center in Manhattan distance – so the maximum propagation distance is still $N$. Thus, in terms of worst-case distance alone, a central prompt or an all-boundary prompt cover the grid in a comparable radius. However, the crucial difference lies in which region of the image suffers the maximum distance (minimal influence). With boundary prompts, it is the central region that is farthest – but we want the center to be least affected (to preserve semantic content). With a central prompt, it is the border region that is farthest – meaning the periphery of the image gets the weakest control. For many vision tasks, the periphery often contains background or less critical details, whereas the center often contains the main subject; therefore, it is far preferable that the center be the least altered region. Boundary placement guarantees this, whereas central placement does the opposite. Furthermore, if we use not one but a distributed set of prompts, the boundary configuration can cover the image more uniformly. In fact, the outer-frame prompt set $B_t$ consists of all $4S_t - 4$ edge positions (for an $S_t \times S_t$ map), surrounding the entire image. Most interior tokens will be very close to some edge (e.g. a token near the top of the image is only a few pixels from the top prompt band), and only the very middle of the image has the maximal distance $N$. By contrast, a small set of prompts in an interior region would leave large areas of the image (e.g. all four corners or sides) far from any prompt. Thus, outermost-frame prompts minimize the area of the feature map that lies at high propagation distance. This yields more uniform and efficient coverage: prompt signals originate from all sides and diffuse inward, reaching a given interior token from multiple directions. We can formalize this advantage by comparing distance distributions. For boundary prompts, the fraction of tokens within a distance $d$ of some prompt grows rapidly with $d$ – indeed, for distance $d < N$ the "uncovered" region is an inner square of side $(S_t - 2d)$, whose area shrinks quadratically as $d$ increases. In contrast, for a centralized prompt, the covered region within distance $d$ is just a central disk/square of area $\sim (2d + 1)^2$, and a large peripheral ring remains uncovered until $d$ approaches $N$. As a result, for any reasonably small $d$, far more of the image is reached by boundary prompts than by an equal number of interior prompts. In a dynamic diffusion sense, if prompt signals propagate outward one step at a time, the boundary

configuration will influence a large portion of the image in the first few propagation steps (with only the middle lagging behind), whereas a central source influences only the vicinity at first and leaves all outskirts unaffected until much later. This spatial coverage analysis underscores the efficiency of having prompts on the outer frame.

It is also useful to consider a more general placement: suppose prompts are placed on some intermediate frame $c$ (neither center nor outermost). The maximum distance to some token will then be $\max c, ; N - c$. There is a natural trade-off here: if $c$ is small (near the boundary), the center is far; if $c$ is large (near the center), the boundary is far. The worst-case is minimized if $c \approx N/2$, which would balance the farthest distance to center and to edge at roughly $N/2$. Indeed, purely from a graph covering perspective, the "middle ring" of an image could theoretically minimize the absolute worst-case distance to all points. However, such a placement means the center is only moderately distant (on the order of $N/2$) from a prompt, implying a much stronger direct prompt effect on the central content than the outermost frame does. In designing MVP, we prioritize protecting the center from corruption over the marginal gain of reducing the overall radius by a small factor. Empirically, even with outermost prompts the maximum distance $N$ is only about half the image size, and we will show that this still provides effective propagation across the image. In fact, because the outer frame uses a greater number of prompt tokens distributed all around, it compensates for a larger radius by multi-directional influence – the center receives some (weak) signal from all sides rather than a strong push from one close prompt. This multi-source arrangement can guide the model subtly and consistently from the periphery, rather than risking a heavy-handed alteration near the center.

**Graph-Theoretic and Diffusion Perspective**   We can cast the above intuitions in terms of graph diffusion or boundary-value problems, which lends another angle to why outer-frame prompting is advantageous. Imagine the feature map as a weighted graph, where each token is a node and edges connect neighboring tokens (e.g. adjacent in the grid). Prompt tokens can be seen as sources introducing a certain state or "potential" into the network, which then spreads to other nodes through the edges. Placing prompt sources on the boundary is analogous to setting boundary conditions in a diffusion process. For instance, one could imagine an iterative update where at each network layer or time step, tokens influence their neighbors (like heat diffusing). If we initialize all boundary nodes with a certain perturbation value and interior nodes at zero, the diffusion or random-walk process will cause the interior to gradually absorb influence from all sides. Classical results for diffusion on a 2D domain tell us that with boundary sources, the interior will be harmonically influenced from all boundaries, often resulting in a smooth gradient that is minimal at the center (the point maximally distant from all sources). By contrast, if a source is at the center, a diffusion process would send a wave of influence outward, with the strongest effect concentrated near the source and decaying toward the boundaries. In a steady state (e.g. solving Laplace's equation with a fixed value at the prompt sources), having the boundary held at a certain perturbation value yields an interior solution that is lowest at the center (the furthest point from the boundary conditions). In other words, the center stays relatively untouched when control signals are applied at the periphery – which is precisely what we want to ensure. This diffusion analogy reinforces that boundary prompts provide a global, gentle influence that permeates the feature map from the outside in, whereas interior prompts act more like local shocks that could disrupt central contents.

From a graph-theoretic view, one can also consider the concept of a dominating set: the prompt set $B$ "dominates" the graph if every node in the graph is within a certain distance $r$ of some prompt. The smallest such $r$ for a given $B$ is the covering radius of that prompt set (our $D_{\max}$ above). The entire outer boundary $B_t$ forms a dominating set with radius $N$. While a smaller dominating radius could be achieved by a carefully chosen subset of interior nodes, those interior nodes would inherently be closer to the center (reducing central distance and increasing corruption risk). The boundary set has the special property that it maximizes the distance to the most sensitive node (the center) while still maintaining a reasonable covering radius. In fact, under the constraint that no prompt is closer than distance $d_{\min}$ to the center, the outermost ring yields the minimal possible covering radius. For example, if we require prompts to be at least $N$ away from center, the only feasible locations are on the outer frame; if we slightly relax to at least $N! - !1$ away, the second-outermost frame becomes available, but using that instead of the outer frame would only shrink the radius by 1 at the heavy cost of moving prompts closer to center. Thus, given the design constraint to maximize center safety, outer-frame placement is the optimal choice for broad coverage.

**Rigorous Justification of Outermost-Frame Optimality**  We now synthesize the above points into a more rigorous justification. Proposition: Placing prompt tokens on the outermost frame simultaneously minimizes the prompt's direct impact on central tokens and provides near-optimal coverage of the entire feature map. More concretely: (a) For any fixed attenuation function $f(d)$ that decreases with $d$, the maximum possible minimal distance between any prompt and the center of the map is achieved by prompts on the boundary — this maximizes the center's distance to the nearest prompt, thereby minimizing the upper bound on prompt-induced change to central features. (b) Subject to (a), the outermost-frame configuration ensures that the distance from any token to some prompt is bounded by $N = \mathcal{O}(S_t)$, and no other configuration with equal or greater center distance can achieve a smaller worst-case distance.

Proof Sketch: (a) is straightforward — the farthest any point can be from the center in an $S_t \times S_t$ grid is along the boundary. Any prompt placed at an interior location has a distance to center $< N$, whereas prompts on the edge have distance $N$ to center; hence the boundary placement uniquely achieves the maximum center distance $N$. Thus, if minimizing $I_{\text{center}} = \delta, f(d_{\text{center-prompt}})$ is paramount, one should choose $d_{\text{center-prompt}}$ as large as possible, i.e. place prompts at the periphery. (b) Now, given that prompts are confined to those at least distance $N$ (or $N - 1$, etc.) from center, we consider the coverage of the map. The outer boundary $B_t$ is one natural choice meeting this constraint. Could any other allowed configuration cover the map more efficiently (i.e. with a smaller maximum distance)? Suppose we remove some subset of boundary prompts or move some prompts off the exact edge inward by one cell. Any interior point that was previously nearest to a removed prompt will now be farther from the remaining prompt set, increasing the worst-case distance. In fact, removing or insetting prompts can only increase the covering radius unless other prompts are moved inward to compensate – but moving any prompt inward violates the center-distance constraint or at least lowers the center distance achieved. The full ring of boundary tokens is a redundant but robust cover – it may use slightly more tokens than minimally necessary for coverage, but this redundancy guarantees no gaps in coverage and uniformly short distances except at the very center. Formally, one can show that $B_t$ (the set of all edge positions) minimizes the function $\Phi(B) = \max_{v \in V} \min_{u \in B} d(u, v)$ among all sets $B$ that satisfy $\min_{u \in B} d(u, \text{center}) = N$. In other words, under the condition that no prompt is closer than $N$ to the center, $B_t$ yields the minimal possible $\Phi(B)$ (which in fact equals $N$ in this case). Any other set that also keeps prompts off the $n < N$ inner frames will either have the same worst-case distance $N$ or worse. Thus, outermost placement is Pareto-optimal in balancing the two objectives: you cannot increase the center safety without also worsening coverage, and vice versa, and the chosen design maximizes center safety while only modestly sacrificing distance-efficiency (staying within a factor of 2 of the absolute minimal radius achievable by any prompt configuration, which is a small price for protecting central content).

In summary, our expanded theoretical analysis confirms that introducing prompts in the outermost square frame is an optimal or at least highly well-founded design choice. It minimizes the risk of core feature corruption by keeping prompts as far as possible from the image's crucial center, while still efficiently diffusing control signals across the entire feature map from the boundaries inward. This justifies the MVP strategy of injecting learnable prompt tokens in the outermost frame at each scale, as it offers strong signal coverage with minimal adverse impact on the learned visual features at the center of the generation.

## E  PROMPT-COMPLEXITY ANALYSIS

At the $t$-th scale of the VAR's patch size set $\mathcal{P}$, let the spatial feature map be of size $S_t \times S_t$. The corresponding **prompt budget** is determined by the number of tokens $\mathcal{N}_t$ introduced at this scale. Below, we analyze the computational overhead associated with different prompt injection strategies.

- **Full Feature-Map Prompt.** Each spatial location is placed as a prompt token:

$$\mathcal{N}_t = \mathcal{N}_t^{\text{full}} = S_t^2.$$

  This leads to a quadratic increase in token count. For example, $S_t = 16 \Rightarrow \mathcal{N}_t = \mathcal{N}_t^{\text{full}} = 256$.

- **Outermost Frame Prompt(** $\mathcal{N}^{\mathcal{B}_t} \leq \tau$ **).** Only the outermost frame positions are used:

$$\mathcal{N}_t = \mathcal{N}^{\mathcal{B}_t} = 4S_t - 4,$$

which scales linearly with $S_t$ (e.g., $S_t = 16 \Rightarrow \mathcal{N}_t = \mathcal{N}^{\mathcal{B}_t} = 60$). This formulation is only valid for $S_t > 1$.

- **L-shaped Corner Prompt** ( $\tau < \mathcal{N}^{\mathcal{B}_t}$). This strategy selects strips of stride width $a$ originating from the four corners of the feature map:

$$\mathcal{N}_t = \mathcal{N}_{\mathcal{C}}^{\mathcal{B}_t} = 8a + 4.$$

For example, with $a = 2$ and $S_t = 16$, we get:

$$\mathcal{N}_t = \mathcal{N}_{\mathcal{C}}^{\mathcal{B}_t} = 20.$$

This configuration is applied only when it does not exceed the corresponding square frame prompt count, i.e., $\mathcal{N}_{\mathcal{C}}^{\mathcal{B}_t} \leq \mathcal{N}^{\mathcal{B}_t}$.

This comparison reveals a clear computational hierarchy:

$$\mathcal{N}_t^{\text{full}} \sim \mathcal{O}(S_t^2) \gg \mathcal{N}^{\mathcal{B}_t} \sim \mathcal{O}(S_t) \gg \mathcal{N}_{\mathcal{C}}^{\mathcal{B}_t} \sim \mathcal{O}(1),$$

where $\mathcal{N}_t^{\text{full}}$, $\mathcal{N}^{\mathcal{B}_t}$, and $\mathcal{N}_{\mathcal{C}}^{\mathcal{B}_t}$ denote the token budget under full feature map, outermost square frame, and l-shaped corner prompting respectively. In practice, the l-shaped corner strategy reduces the token cost by a factor of $\sim 5$–$10\times$ compared to border prompting, and over $\sim 30\times$–$100\times$ compared to dense prompting, while retaining sufficient spatial coverage. This establishes an efficient trade-off between semantic guidance fidelity and computational complexity.

## F  LIMITATION

While MVP demonstrates good performance in class-to-image and text-to-image generation, there are still some limitations that warrant further investigation.

**Limitation of CLIP's text encoder.** We employ only the default CLIP text encoder throughout all experiments. While more powerful or specialized language encoders may potentially yield better generation quality, particularly in text-to-image tasks, we did not pursue this direction due to the principle of minimizing model modifications in prompt tuning.

**Limitation of Visual-Quality and Semantic-Quality Balance.** Frankly speaking, compared to the significant improvements in metrics like IS after applying MVP, the improvement in FID, a low-level fidelity metric, is limited. This can be attributed to the fact that our method introduces prompts at the outermost frames of feature maps, thereby primarily emphasizing semantic information. As a result, it inherently favors metrics like IS that capture cross-category semantics. In contrast, the improvement in FID typically requires optimization in the pixel space. While introducing perturbations for each input token in the pixel space could effectively improve FID, it would also bring substantial computational cost, which contradicts the original intention of prompt tuning. In conclusion, this is an inherent limitation of prompt tuning.

# G   ALGORITHM OF MVP

---

**Algorithm 1:** Multi-Scale Visual Prompt

---

**Input:** $\mathcal{P} = \{S_1, \ldots, S_T\}, \tau$

**Output:** $\mathcal{V}^{\mathcal{B}}, \mathcal{I}_{\mathrm{id}}$

$\mathcal{V}^{\mathcal{B}} \leftarrow Queue(), \mathcal{I}_{\mathrm{id}} \leftarrow Queue()$ ; // ▷ initialize prompt and indices queues

**forall** $S_t \in \mathcal{P}$ **do**

   $\mathcal{N}^t \leftarrow 4S_t - 4$ ;                   // ▷ compute number of outermost tokens

   **if** $\mathcal{N}^t \leq \tau$ **then**

      $\mathcal{N}^{\mathcal{B}_t} \leftarrow \mathcal{N}^t$ ;            // ▷ set prompt count to outermost size

      $\mathcal{I}_{\mathrm{id}}^{\mathcal{B}_t} \leftarrow Outermost(S_t)$ ;      // ▷ select outermost token indices

   **else**

      $a \leftarrow \left\lfloor \frac{\tau - 4}{8} \right\rfloor$ ;     // ▷ compute stride width for corner sampling

      $\mathcal{N}^{\mathcal{B}_t} \leftarrow 8a + 4$ ;      // ▷ set prompt count under threshold

      $\mathcal{I}_{\mathrm{id}}^{\mathcal{B}_t} \leftarrow LCorner(S_t, a)$ ;     // ▷ select L-shaped corner indices

   $\mathcal{V}^{\mathcal{B}_t} \leftarrow Visual\_Prompts(\mathcal{N}^{\mathcal{B}_t})$ ;

   Queue\_Push$(\mathcal{I}_{\mathrm{id}}, \mathcal{I}_{\mathrm{id}}^{\mathcal{B}_t})$ ;          // ▷ enqueue selected indices

   Queue\_Push$(\mathcal{V}^{\mathcal{B}}, \mathcal{V}^{\mathcal{B}_t})$ ;          // ▷ enqueue generated prompts

**return** $\mathcal{V}^{\mathcal{B}}, \mathcal{I}_{\mathrm{id}}$

---

## H  VISUALIZATION

Figure 6: Visualization of class-to-image samples generated using MVP (512×512).

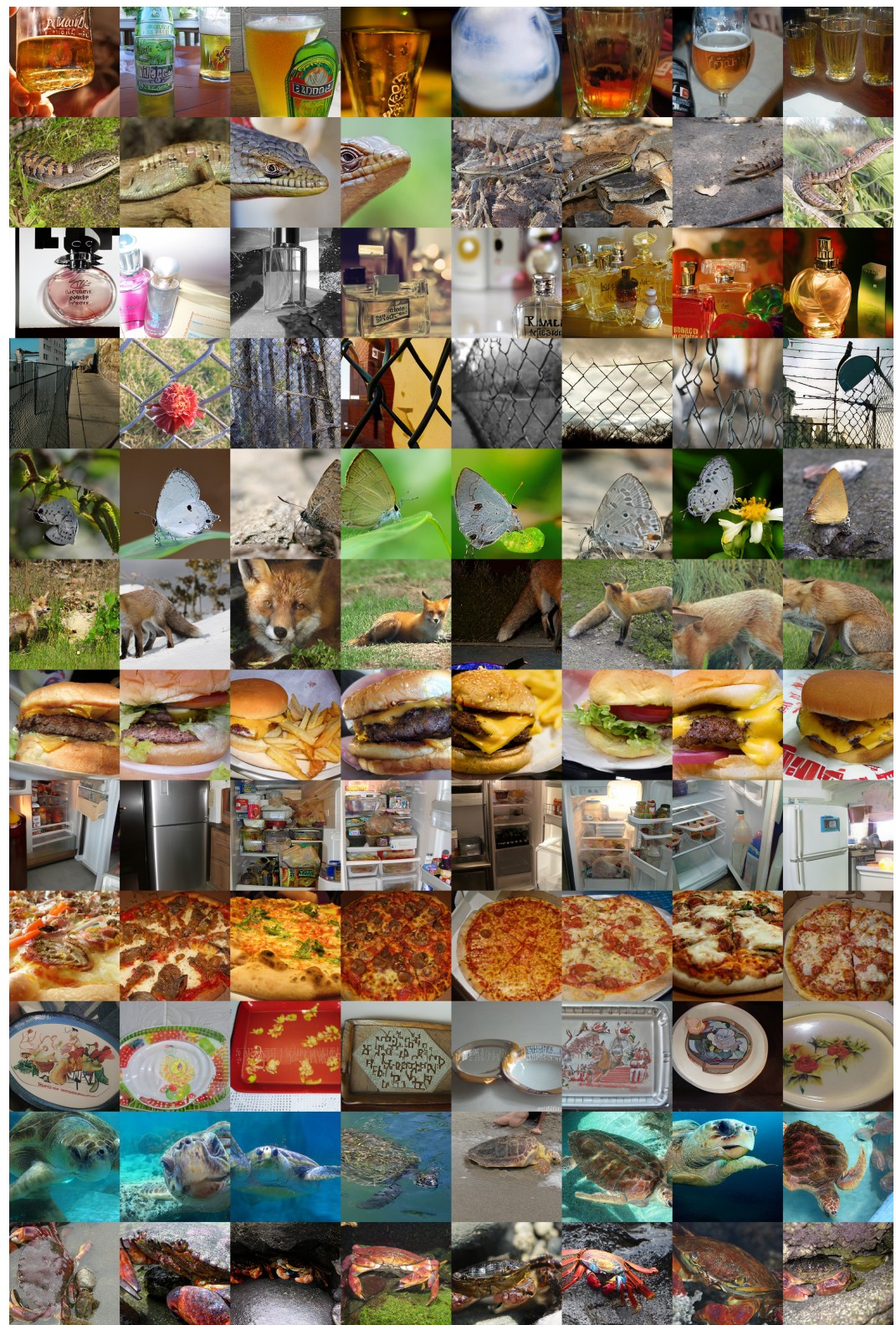

Figure 7: Visualization of class-to-image samples generated using MVP (256×256).

MVP                                           VAR

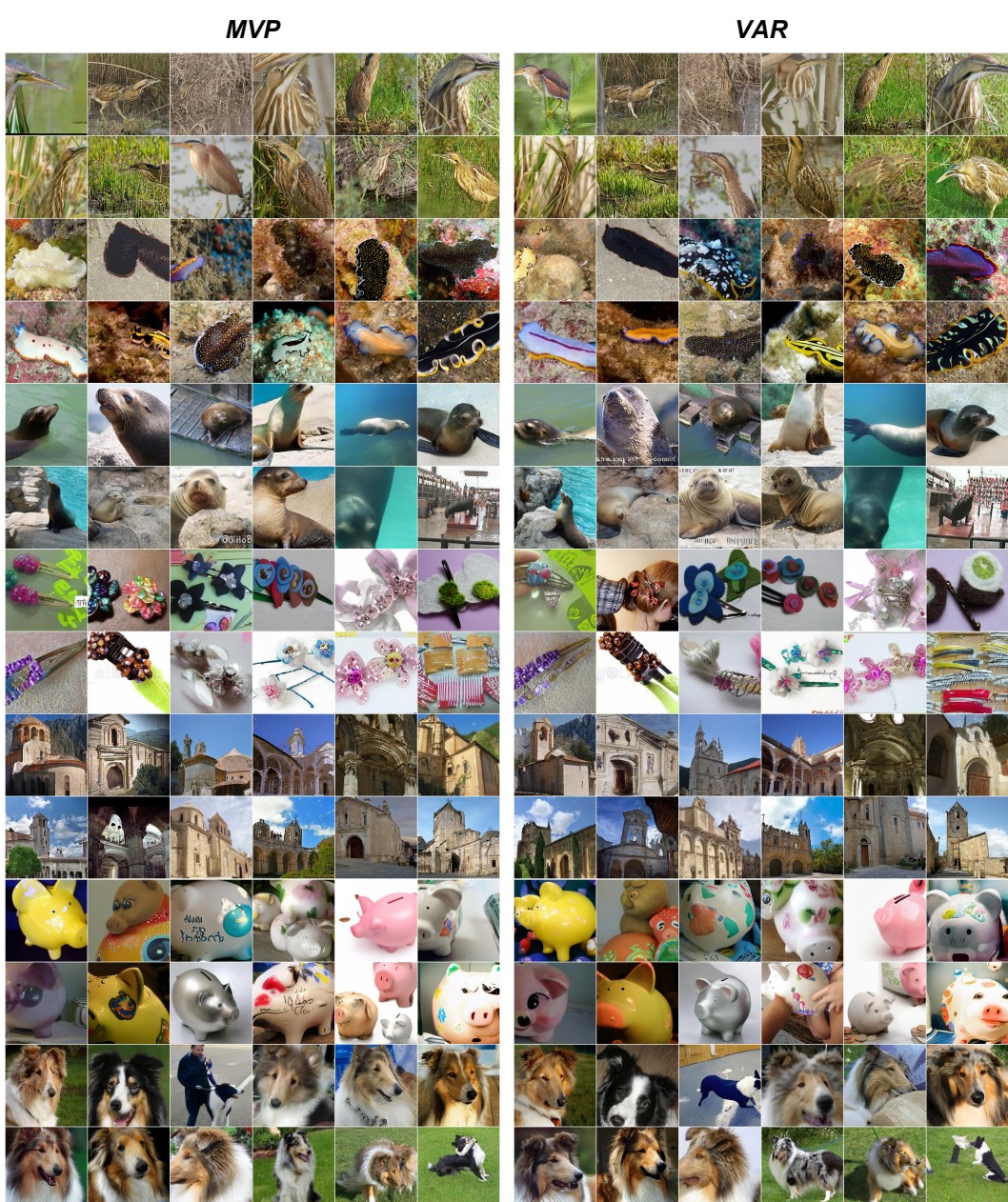

Figure 8: Comparison of class-conditional image samples generated by MVP and VAR at 256×256 resolution.

**Class-to-Image.**    Visualization of samples conditioned on class labels. A subset of class labels is randomly sampled from the validation set, and multiple images are generated for each class using MVP. Each row corresponds to one class, and columns show diverse samples under the same condition. Fig. 6 and Fig. 7 present the results at 512×512 and 256×256 resolutions, respectively. As shown in Fig. 8, compared to VAR, our method produces more detailed and semantically accurate generations, demonstrating stronger alignment with the class-conditional guidance.

**Text-to-Image.**    Fig. 9 and Fig. 10 illustrate examples of text-to-image generation results produced by MVP at 256 × 256 resolution. Although VAR is originally trained for class-conditional generation, these results demonstrate that it can be readily adapted to free-form text prompts through minimal finetuning with paired image-text data. The generated samples exhibit diverse visual con-

tent and strong semantic consistency with the input text, indicating MVP ability to generalize beyond class supervision with minimal adaptation.

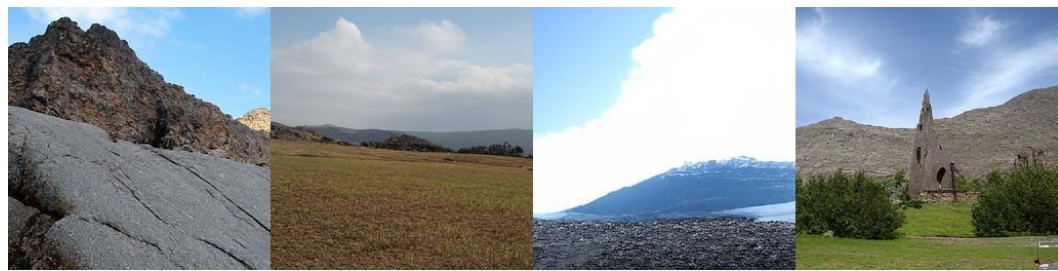

a mountain under a blue sky.

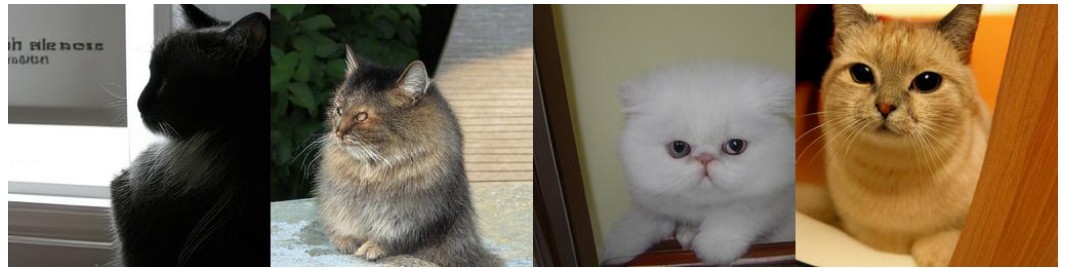

a photo of little cute cat.

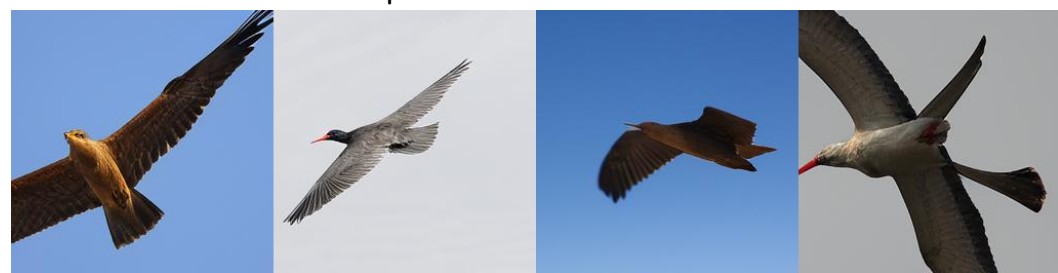

a bird flying in the sky.

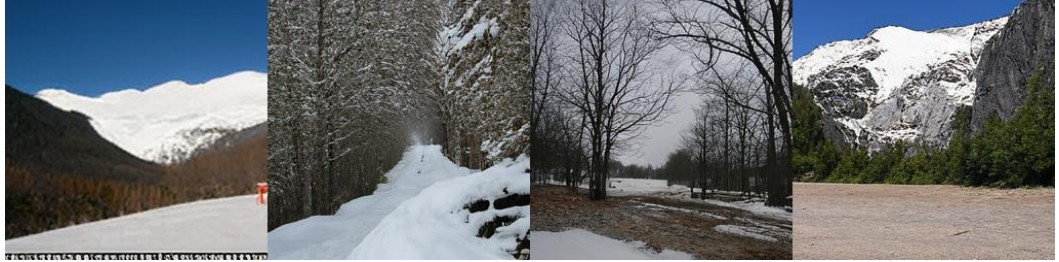

a snowy road in a forest.

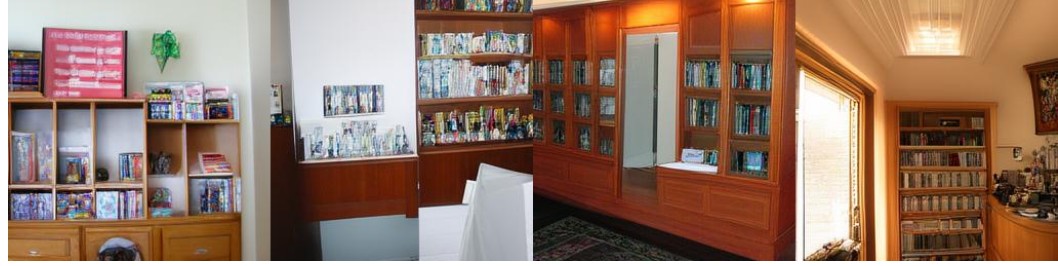

a bookshelf full of books.

Figure 9: Visualization of text-to-image samples generated using MVP ($256 \times 256$).

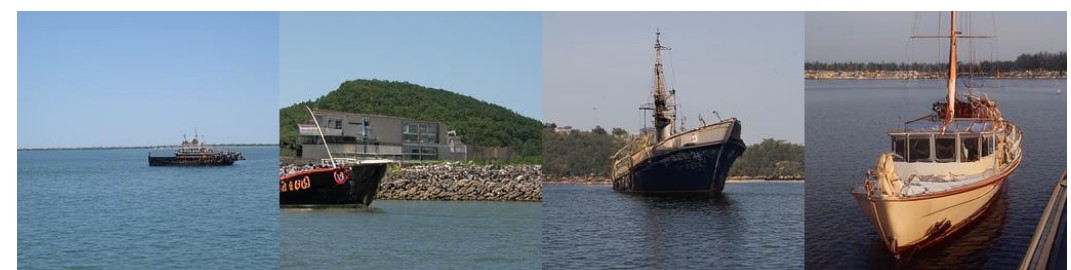

a boat on a calm lake.

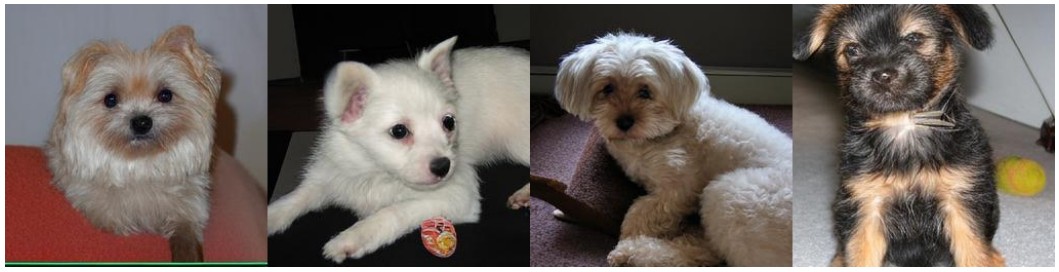

a small dog lying on a sofa.

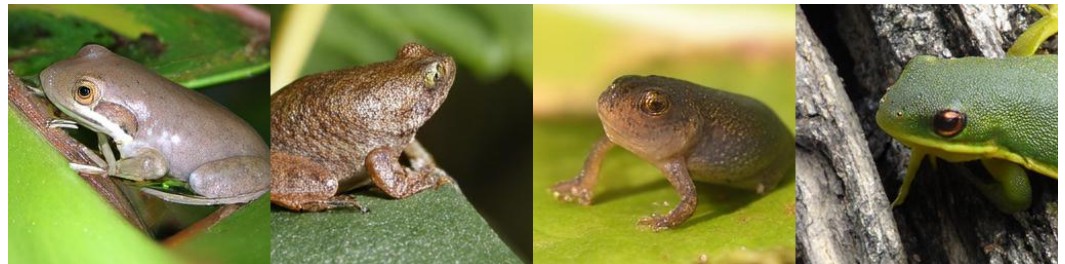

a frog sitting on a leaf.

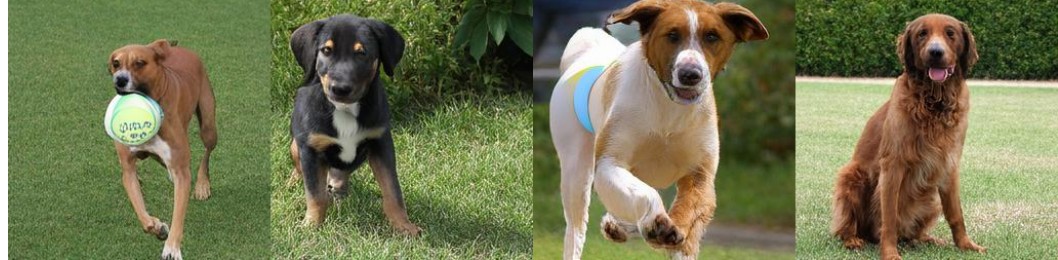

a dog play on the grass.

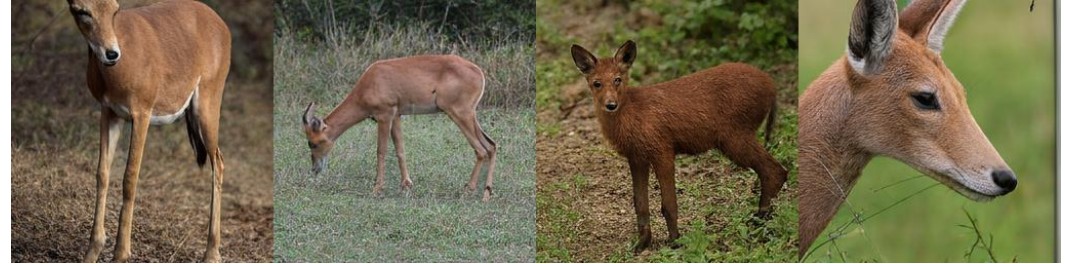

a deer standing in the woods.

Figure 10: Visualization of text-to-image samples generated using MVP ($256 \times 256$).