# OpenReview forum: "MVP: Multi-scale Visual Prompt for Visual AutoRegressive Generation"
_ICLR.cc/2026/Conference — Submitted to ICLR 2026_

### Official Review · Reviewer_o6w3 · 2025-10-27

**Soundness:** 2
**Presentation:** 2
**Contribution:** 3
**Rating:** 4
**Confidence:** 3

**Summary:**

This paper introduces MVP, a perturbation-based prompt tuning method, for the VAR models. Instead of adding embeddings, the authors propose to add learnable perturbation tokens on the outermost square frame of the feature map, which they claim it can minimize impact on the image center, and thereby avoiding model feature corruption (line 184). To resolve the lacks of semantics which traditional perturbation prompt tuning has, the authors use multi-level semantic refinement (line 271) with multi-level CLIP contrastive loss (line 287-291), which enables the class-to-image improvements (Table 1) and extending VAR to text-to-image with much less trainable parameters (Table 3) and GPU computation time (Table 4). For the experiments, author has shown MVP improving FID/IS over VAR at 256 x 256 (line 84) and 512 x 512 (line 326) on ImageNet.

**Strengths:**

$\bullet$ Simple and Effective method: only adding prompts on the outermost frame is simple, which requires minimal architectural modifications.

$\bullet$ Convincing statistics: MVP achieves better performance than VAR's with less than one percent extra parameters (line 325) and 0.54% of VAR-CLIP training GPU-hours (line 86). Plus, MVP presents better performance over VAR at 256 x 256 and 512 x 512 on ImageNet.

$\bullet$ Interesting Mutli-level semantic refinement: class $\to$ sentence  $\to$ caption CLIP supervision aligned to different generation stages is well-motivated.

**Weaknesses:**

$\bullet$ Lack of Design Justification: Section 3.1.2 seems to claim that the prompt should live on the outermost frame because a perturbation’s effect on the center decays with distance. Then, the authors uses an attenuation factor $\alpha$ to derive that if a perturbation is placed far away (frame 0), it will impact the center less than the perturbation placed closer (frame c). $Impact(n \to N) = \delta \cdot \alpha^{N-n} \Rightarrow I_0 < I_c$. This is a overly simplified model of propagation which depends only on distance, and I think authors need to come up a full causal analysis of the actual VAR stack in order to validate the claim. Current result lacks broad ablations against other spatial layouts.

$\bullet$ On Line 394, authors claim first-scale prompting hurts class-to-image but helps text-to-image. I think authors need to explain why it hurts class-to-image and providing some diagnostics.

$\bullet$ The current text-to-image baselines and metrics are limited, which mainly focused on IS/FID, sparse human or robustness checks. Moreover, the generality beyond VAR backbones is under-tested.

$\bullet$ Lack of Clarity in the paper presentation and organization:
In figure 1, there is a typo, and it should be "framework". In Section 3.1.2, authors defines $\alpha$ as the signal attenuation factor, but on Line 277, $\alpha$ is redefined as a hyperparameter in $(0,1)$. This is an example of abuse of notations. On Line 182, do authors mean to use "denoted" instead of "donated"? I recommend the authors to check the rest of the paper to enhance clarity.


There are some chances that I misunderstood some parts of this paper, and I welcome corrections and an active discussion from the authors.

**Questions:**

$\bullet$ Author's current attenuation model $Impact(n \to N)$ assumes distance-only decay. Can author provide me more empirical evidence that shows the real VAR stacks follow this decay? And the center corruption is minimized by the outermost frame?

$\bullet$ Does the optimal layout change with scale depth, token budget $\tau$, or resolution?

$\bullet$ Why is the outermost frame optimize versus other spatial layouts? Please provide ablations.

$\bullet$ Regarding the Weakness 2, Can you explain why first-scale prompts degrade class-to-image but helps text-to-image? Please provide the diagnostics that shows the degradation.

$\bullet$ You claim that VAR and "VAR-like" models serve as the target model for MVP (line 142). Can you port MVP to at least one masked-AR and one encoder-decoder image generator to show the architecture agnostic benefits?

---

> ### Author Response · Authors · 2025-11-25
> **Part 1**
>
> Hi Reviewer `o6w3`, we sincerely appreciate you for taking the time to review our work and for providing both encouraging and constructive feedback. We highly value your comments and provide our detailed responses below. Should you have further questions or wish to discuss any part of our work in more depth, we would be very glad to continue the conversation.
>
> >**W2 & Q4: Why first-scale prompt hurts class-to-image task.**
>
> We sincerely thank the reviewer for bringing attention to this interesting phenomenon, and are happy to provide a detailed explanation. We would also like to clarify that this behavior does not occur in all C2I settings, but only under specific conditions.
>
> In our experiments, we observed that when doing the task of improving VAR’s class-conditional generation performance on ImageNet ($256 \times 256$), injecting prompts at the first scale indeed degrades generation quality. However, this effect does not appear in other tasks—such as the transfer experiments in Table 8—where first-scale prompting remains stable and does not cause any degradation. As the results show below:
>
> | **Method** | **Mean(FID ↓)** | **SUN397(FID ↓)** | **Food101(FID ↓)** | **Resisc(FID ↓)** |
> |------------|----------|------------|-------------|------------|
> | MVP(Ours) | 29.72 | 22.42 | 28.76 | 37.97 |
> | MVP(w/o 1st scale prompts) | 29.81 | 22.46 | 28.91 | 38.05 |
>
> We believe the key reason arises from both the architectural specificity of VAR and the task-specific characteristics of the setting discussed at Line 394(improving VAR’s class-to-image generation quality):
>
> ***1. Architectural Specificity of VAR.*** The first-scale input to VAR is the **class embedding**. In the VAR-backbone, this class embeddings matrix has already been fully trained and is highly optimized. Injecting prompt tokens at this scale introduces perturbations directly into the backbone’s class-conditioning interface. Even very small perturbations can shift or distort the pretrained class embedding, causing the conditioning vector to destabilize and consequently harming the overall generation quality.
>
> ***2. Transfer to New Data Distributions.*** In transfer tasks such as those shown in Table 8, the model no longer relies on the VAR's pretrained class embeddings matrix. Instead, the class embeddings representation must be adapted to a new data distribution. In this scenario, first-scale prompting does not corrupt a well-learned embedding; rather, it assists the fast adaptation of the conditioning representation. As a result, the negative effect seen on ImageNet does not appear, and the advantages of prompt tuning become clearly evident.
>
> We believe that the above clarifications and supporting experiments sufficiently alleviate the reviewer’s question.

---

> ### Author Response · Authors · 2025-11-25
> **Part 2**
>
> >**W3: The current T2I metrics are limited, and the generality beyond VAR backbones remains under-tested.**
>
> We sincerely thank the reviewer for raising this point. While we acknowledge that, for general text-to-image systems, existing baselines and metrics are often limited. However, after a detailed analysis within the scope and nature of our experiments, we found that the current evaluation protocol is already adequate. In particular, additional T2I metrics (such as GenEval) are not well aligned with our task formulation and experimental setup, mainly for the following reasons:
>
> ***1.Experimental positioning.*** The T2I experiment in our paper is positioned as an exploratory extension of VAR (similar in spirit to VAR-CLIP[r5]), rather than a fully developed text-to-image model (such as HART [r2], STAR [r4], and Infinity [r1]). Our goal is to study whether MVP can effectively extend VAR with text conditioning under a constrained setting, not to build or benchmark a dedicated large-scale T2I model.
>
> ***2.Limited text encoder capability.*** Given the exploratory positioning of our extension, our experimental setting deliberately adopts a lightweight design: we only use a basic CLIP text encoder to enrich the conditional embedding when transitioning from class labels to text inputs, rather than relying on a powerful large language model or a heavily optimized text encoder typically used in modern large-scale T2I models. Thus, it is not equipped with strong text–image alignment capacity by design.
>
> ***3.Limitations of data scale and quality.*** State-of-the-art autoregressive T2I models (such as HART, STAR and Infinity) are trained on large-scale, high-quality, often private datasets with hundreds of millions of text–image pairs. In contrast, our T2I experiment is trained only on a small public dataset and is intended as an exploratory extension study, rather than a full-scale text-to-image benchmark. As a result, many advanced T2I evaluation metrics designed and calibrated for large models trained on massive, high-quality datasets—tend to become unstable or uninformative in such a low-data exploratory setting.
>
> Consequently, we focus on FID/IS and other metrics reported in the paper, and position the T2I results as an interesting and encouraging byproduct of extending VAR with MVP, rather than as a key contribution. Notably, the empirical results further support this positioning: with less than 1% of the compute cost used in
> , MVP's prompt tuning based method is already sufficient to endow the VAR backbone with basic yet effective text-to-image generation ability and achieved competitive results compared with VAR-CLIP. These results show that even with simple prompt tuning on top of VAR, our method can already yield surprisingly competitive generations under this constrained setting.
>
> We again thank the reviewer for this constructive suggestion and fully acknowledge the importance of evaluating more representative text-to-image baselines. We also commit to conducting more comprehensive T2I studies once higher-quality and larger-scale datasets become available to us. In parallel, regarding your concern about the limited baselines, our appendix already includes text-conditioned experiments based on two widely adopted and stronger VAR-like models—Infinity[r1] and HART[r2]—which provide a more reliable evaluation of MVP’s behavior under text-guided generation.
>
> Regarding your suggestion to further strengthen the evidence for MVP’s generality beyond class-conditional settings, we additionally evaluate an open-domain super-resolution task built upon a VAR-based framework. Considering that HART adopts a more flexible rotary positional encoding than VAR and provides a stronger architectural backbone for spatially aligned prediction, we integrate MVP into the HART framework and compare it against VARSR[r3] as the baseline. This setup allows us to more fairly assess whether MVP can improve open-domain super-resolution performance while avoiding limitations inherited from the original VAR architecture. The evaluation results under the DRealSR real-world benchmark are as follows:
>
> | **Model** | **PNSR ↑** | **SSIM ↑** | **LPIPS ↓** | **FID ↓** |
> |-----------|------------|----------| ----------|---------|
> | VARSR(baseline) | 28.16 | 0.7652 | 0.3541 | 155.87 |
> | HART + MVP (ours) | 27.67 | 0.7701 | 0.3626 | 151.38 |
>
> These results demonstrate that MVP is compatible with open-domain restoration pipelines and achieves competitive performance on such tasks, providing stronger evidence that MVP can generalize beyond class-conditional generation and beyond the standard VAR backbone.
>
> We believe that the above clarifications and supporting experiment sufficiently alleviate the reviewer’s concerns.

---

> ### Author Response · Authors · 2025-11-25
> **Part 3**
>
> >**W4: Concerns about the clarity of presentation and organization.**
>
> We sincerely thank the reviewer for the time and care taken to point out the clarity and notation issues present in our manuscript. We acknowledge these issues and sincerely apologize for the lack of polish. We have corrected all mentioned problems and conducted a comprehensive proofreading pass to improve the clarity and presentation quality of the revised paper. For example, during the proofreading, we identified and fixed the typo in Algorithm 1, where $S_t$ was mistakenly written as $S_k$. All related notation has now been made fully consistent.
>
> &nbsp;
>
> >**Q1: Provide more empirical evidence that real VAR stacks exhibit this decay and that outermost prompting minimizes center corruption.**
>
> We respectfully thank the reviewer for raising this important question. Below we show that our empirical results already provide direct evidence for both (i) the decay behavior across real VAR stacks and (ii) the minimal center corruption achieved by the outermost-frame strategy.
>
> ***1.Evidence that real VAR stacks follow the decay behavior.*** Our ablation experiments in Table A (different depth) and Table B (different token budgets) reveal a clear and consistent pattern: the monotonic improvement as prompts move outward directly reflects the spatial sensitivity decay of real VAR stacks: center > mid-inner > outer in vulnerability. This provides strong empirical confirmation of the distance-based decay hypothesis raised in the reviewer’s question.
>
> ***2.Evidence that Outermost-Frame minimizes center corruption.*** Across all tested depths and token budgets, the outermost-frame placement achieves the best generation quality. This strongly suggests that:(i)Prompts placed too close to the center disturb the scale-level causal chain;(ii)Outermost prompts influence the backbone minimally, preserving spatial coherence in the central region.
>
> We believe that the above clarifications sufficiently alleviate the reviewer’s concerns.
>
> >**Q2: Does the optimal layout change with scale depth, token budget and resolution?**
>
> We sincerely thank the reviewer for this insightful question. Below we provide a detailed clarification regarding the stability of the optimal outermost-frame layout across scale depth, token budget, and resolution.
>
>
> ***1. Scale depth.*** The results in Table A directly address the depth. We evaluate all four alternative placement strategies under various depth setting(16, 30). The results show that the outermost-frame consistently achieves the best FID/IS. This indicates that increasing the number of scales does not change the relative effectiveness of our spatial layout.
>
> ***2. Token budget.*** To further examine whether the optimal layout depends on the prompt token budget, we additionally conduct experiments on the Depth-30 VAR model with different token budgets while keeping all layout variants identical to those used in Table A. The results in Table B show that our outermost-frame design achieve the best performance across different token budgets. This further demonstrates that the optimality of our placement strategy is robust to changes in token budget and is not tied to a specific prompt token capacity.
>
> **Table B. Ablation on prompt position designs under different token budgets. '*' indicates mean results over 3 runs with different random seeds**
>
> | **Token budget** | **Placement Strategy** | **FID ↓** | **IS ↑** |
> |-----------|-------------------------|-----------|-----------|
> | 20        | **Ours (Outermost-Frame)** | **2.12** | **281.6** |
> | 20        | Random | 2.16* | 273.7* |
> | 20        | Innermost  | 2.15 | 270.6  |
> | 20        | Center-to-Outer | 2.15 | 275.8 |
> | 20        | Sub-Outermost | 2.13 | 279.1 |
> | 28        | **Ours (Outermost-Frame)** | **2.03** | **292.9** |
> | 28        | Random | 2.13* | 279.4* |
> | 28        | Innermost  | 2.17 | 278.1  |
> | 28        | Center-to-Outer | 2.12 | 286.5 |
> | 28        | Sub-Outermost | 2.05 | 289.4 |

---

> ### Author Response · Authors · 2025-11-25
> **Part 4**
>
> ***3. Resolution.*** To examine whether the optimal layout remains consistent under different resolutions, we further conduct experiments using a VAR backbone (depth=36) trained at a higher resolution of 512×512 (Table C). The results in Table C show that even at a substantially higher resolution, our outermost-frame design continues to achieve the best performance across all prompt-position designs. This demonstrates that the optimality of our design is resolution-invariant and remains stable when scaling to higher-resolution generation.
>
> **Table C. Ablation on prompt position designs for improving VAR’s class-conditional generation performance on ImageNet ($512 \times 512$). '*' indicates mean results over 3 runs with different random seeds**
>
> | **Placement Strategy** | **FID ↓** | **IS ↑** |
> |-------------------------|-----------|-----------|
> | **Ours (Outermost-Frame)** | **2.47** | **317.4** |
> | Random | 2.55* | 308.2* |
> | Innermost  | 2.57 | 298.6  |
> | Center-to-Outer | 2.51 | 300.5 |
> | Sub-Outermost | 2.49 | 312.9 |
>
> These experiments collectively demonstrate that the optimal layout remains stable across changes in scale depth, token budget, and resolution. We believe that the above supporting experiments sufficiently alleviate the reviewer’s concerns
>
> &nbsp;
>
> >**Q5: Can you port MVP to at least one masked-AR and one encoder-decoder image generator to show the architecture agnostic benefits?**
>
> We sincerely thank the reviewer for the suggestion. However, we respectfully note that this question appears to fall outside the scope of our work. MVP is not proposed as an architecture-agnostic method; rather, it is deliberately designed for VAR and VAR-like autoregressive models that operate with next-scale prediction instead of next-token prediction. Accordingly, evaluating MVP on masked-AR or encoder–decoder generators does not align with the intended formulation of our approach.
>
> We believe the reviewer is aware of this design scope, as explicitly stated in the first sentence of your summary: “a perturbation-based prompt tuning method for the VAR models.” Therefore, directly porting MVP to masked-AR or encoder–decoder frameworks is beyond the intended design scope of this work.
>
> Nonetheless, we are glad to provide further clarification. Specifically, MVP relies on two key design principles that are tightly coupled with VAR:
>
> ***1. Outermost-Frame Prompt Placement is spatially aligned with scale-level models.*** Our outermost-frame prompt placement strategy is designed specifically for VAR and VAR-like models, as it inherently leverages their scale-level parallel modeling characteristics. In masked-AR (e.g., MaskGIT) , token selection follows random masking order. When a prompt token falls into a masked position, it is ignored by the model, making the spatial prompt ineffective or unstable, thereby defeating the core idea of scale-level guidance.
>
> ***2.Multi-Level Prompt Injection depends on the next-scale prediction mechanism in VAR.*** MVP progressively injects contextual prompts using multi-stage textual tuning strategy, providing hierarchical guidance throughout the multi-scale generation process. In contrast, encoder-decoder models (e.g., diffusion, VQ-GAN) do not include explicit multi-scale stages, and therefore do not provide the hierarchical conditioning points required for MVP to function as designed.
>
>
> We believe that the above explanations sufficiently address the reviewer’s question and clarify why MVP is intentionally designed for VAR and VAR-like models rather than as an architecture-agnostic method.
>
> &nbsp;
>
> &nbsp;
>
> Best wishes and regards,
>
> All authors of Submission 769.
>
> &nbsp;
>
> &nbsp;
>
> &nbsp;
>
> &nbsp;
>
> Reference:
>
> [r1] Infinity: Scaling bitwise autoregressive modeling for high-resolution image synthesis, CVPR2025
>
> [r2] Hart: Efficient visual generation with hybrid autoregressive transformer, ICLR 2025
>
> [r3] Visual autoregressive modeling for image super-resolution, ICML 2025
>
> [r4] STAR: Scale-wise Text-conditioned AutoRegressive image generation, arxiv 2024
>
> [r5] VAR-CLIP: Text-to-Image Generator with Visual Auto-Regressive Modeling, arxiv 2024

---

### Official Review · Reviewer_iowk · 2025-10-31

**Soundness:** 3
**Presentation:** 2
**Contribution:** 2
**Rating:** 4
**Confidence:** 3

**Summary:**

This paper introduces MVP (Multi-scale Visual Prompt), a perturbation-based prompt tuning method designed for visual autoregressive (VAR) models that utilize a next-scale prediction mechanism. The core idea is the "square frame prompt," which introduces learnable prompt tokens exclusively to the outermost border of the feature map at each generative scale. This design aims to achieve efficient information propagation while minimizing feature corruption in the image's center. The authors propose a multi-level semantic refinement strategy for training, using increasingly detailed texts (class labels, sentences, captions) and a CLIP-based contrastive loss to supervise prompt learning. The paper demonstrates that MVP improves the class-to-image generation quality of VAR models and, with minimal computational cost, extends their capability to text-to-image synthesis.

**Strengths:**

- The method is exceptionally efficient. For text-to-image generation, MVP achieves competitive performance using only 0.54% of the training GPU-hours and 0.46% of the trainable parameters compared to a fully trained VAR-CLIP model, making it highly practical.
- MVP demonstrates consistent and notable improvements in generation quality over strong VAR baselines. On ImageNet, it improves both FID and IS scores for class-to-image generation at $256 \times 256$ and $512 \times 512$ resolutions.

**Weaknesses:**

- The primary weakness is the insufficient justification for the core design choice of the "square frame prompt." The paper lacks a crucial ablation study comparing this specific spatial arrangement to other prompt distributions with an equal number of parameters (e.g., random placement, a sparse grid, or prompts concentrated in the center). Without this, it is unclear if the benefits come from the specific "frame" structure or simply from adding learnable parameters at the periphery.
- The theoretical motivation, particularly the concept of the "planar concept," is vague and not rigorously defined. The connection between this abstract idea and the concrete implementation of a square frame prompt feels tenuous and more like a post-hoc rationalization than a guiding design principle.
- The method introduces a key hyperparameter, the threshold $\tau$, which determines when the prompt shape changes to preserve efficiency. The ablation study in Table 6 shows that the optimal value for $\tau$ is sensitive to model depth and lacks a clear pattern, suggesting it requires expensive, model-specific tuning.
- Using VAR to do T2I seems not proper and lack of convenience.

**Questions:**

NA

---

> ### Author Response · Authors · 2025-11-25
> **Part 1**
>
> Hi Reviewer `iowk`, we sincerely appreciate you for taking the time to review our work and for providing both encouraging and constructive feedback. We highly value your comments and provide our detailed responses below. Should you have further questions or wish to discuss any part of our work in more depth, we would be very glad to continue the conversation.
>
> >**W2: Theoretical Motivation**
>
> Thank you for your question. We included the theoretical analysis in our paper for the reason of hoping to fill the current theoretical gap underlying this design. Our exploration is grounded in empirical observations and existing works [r1-r7], and we have also made efforts to provide additional theoretical analysis for this design, which necessarily requires connecting practical findings with theoretical reasoning.
>
> No scientific exploration can be guided by correct principles beforehand; it requires analysis and conclusion-drawing after the fact. Having provided such extensive experiments and analysis, we respectfully disagree with your assessment of post-hoc rationalisation.
>
>
> &nbsp;
>
> >**W3: The threshold is sensitive to model depth and lacks clear pattern, suggesting it requires expensize, model-specific tuning.**
>
> We thank you, but respectfully disagree with the concern about the role of the threshold in determining the efficiency of our method. Our clarification proceeds from two perspectives:
>
> ***1. The threshold actually not only follows a simple and consistent structural rule, but also not sensitive.*** Although the threshold values in Table 6 may appear irregular (4, 12, 20, 28), they actually follow a simple underlying structure of the form $\tau = 8i+4$ based on our outermost square frame placement strategy. More importantly, across all model depths, the common choice of i=2 and $\tau=20$ consistently yields strong performance without requiring depth-specific adjustments. This demonstrates that the threshold is not unstable but instead follows a predictable pattern, and a single default setting ($\tau \ge 20$) works reliably across depths.
>
> ***2. The tuning cost is extremely small compared to other hyperparameter which using in full-parameter tuning.*** To intuitively show the efficiency of our method's tuning cost, we visualize the training loss curves of MVP prompt tuning and VAR full-parameter fine-tuning with respect to training steps under the experimental setup of Table 8 (Comparison with Other PEFT Methods). The visualization results preseted in Appendix D.2 (first paragraph, highlighted blue), MVP exhibits substantially faster convergence and consistently lower loss throughout training compared to VAR across all three dataset (SUN397, Food101, and Resisc). The results show that the additional computation required by MVP's prompt budget tuning is extremely lightweight. Compared with common hyperparameter tuning costs in generative modeling (e.g., learning rate schedules, batch-size), this overhead is minimal and practically negligible. Thus, our prompt budget threshold tuning introduces almost no extra training burden or implementation complexity.
>
> We believe that the above clarifications and supporting evidence sufficiently alleviate the reviewer’s concerns.

---

> ### Author Response · Authors · 2025-11-25
> **Part 2**
>
> >**W4: Using VAR to do T2I seems not proper and lack of convenience.**
>
> We thank the reviewer for raising this point. However, we respectfully do not consider this a weakness.
>
> We agree that using VAR for text-to-image(T2I) generation is not as convenient or straightforward as employing models explicitly designed for T2I (e.g., Infinity[r8], HART[r9], STAR[r10]). We would like to respectfully clarify that this experiment is not intended to position VAR as a practical T2I backbone, but rather to study the feasibility of extending VAR with textual conditioning under a constrained and exploratory setting.
>
> Specifically, developing a fully functional VAR-based T2I model would require substantial additional engineering efforts, including restructuring the model architecture, integrating a strong language encoder, and conducting large-scale training on high-quality text–image datasets. In contrast, our experimental setup is intentionally designed to explore the potential of MVP in extending VAR-based models, rather than to build a complete T2I system. We follow a lightweight setting similar to VAR-CLIP[r11], aiming to investigate whether MVP can effectively leverage textual conditioning to enhance VAR under limited resources and constrained environments. Interestingly, the empirical results reinforce this positioning: despite using less than 1% of the compute cost required by VAR-CLIP, MVP’s lightweight prompt-tuning strategy already equips VAR with basic yet reasonably effective T2I capability, achieving competitive performance relative to VAR-CLIP.
>
> Overall, we believe that the limitations of this T2I setup do not undermine our contributions. On the contrary, they further highlight the robustness of our approach. The results demonstrate that our method enhances VAR’s extensibility to additional conditioning modalities, even under non-ideal backbone settings.
>
> &nbsp;
>
> &nbsp;
>
> Best wishes and regards,
>
> All authors of Submission 769.
>
> &nbsp;
>
> &nbsp;
>
> &nbsp;
>
> &nbsp;
>
> Reference:
>
> [r1] Paint-by-Example: Exemplar-based Image Editing with Diffusion Models, CVPR 2023
>
> [r2] Drag Your GAN: Interactive Point-based Manipulation on the Generative Image Manifold, SIGGRAPH 2023
>
> [r3] Boxdiff: Text-to-image synthesis with training-free box-constrained diffusion, CVPR 2023
>
> [r4] Exploring the Transferability of Visual Prompting for Multimodal  Large Language Models, CVPR 2024
>
> [r5] Exploring Visual Prompts for Adapting Large-Scale Models, arxiv 2022
>
> [r6] Unleashing the Power of Visual Prompting At the Pixel Level, TMLR 2024
>
> [r7] AutoVP: An Automated Visual Prompting Framework and Benchmark, ICLR 2024
>
> [r8] Infinity: Scaling bitwise autoregressive modeling for high-resolution image synthesis, CVPR2025
>
> [r9] Hart: Efficient visual generation with hybrid autoregressive transformer, ICLR 2025
>
> [r10] STAR: Scale-wise Text-conditioned AutoRegressive image generation, arxiv 2024
>
> [r11] VAR-CLIP: Text-to-Image Generator with Visual Auto-Regressive Modeling, arxiv 2024

---

### Official Review · Reviewer_LEkF · 2025-10-31

**Soundness:** 3
**Presentation:** 3
**Contribution:** 3
**Rating:** 6
**Confidence:** 3

**Summary:**

The paper proposes MVP (Multi-scale Visual Prompt), a multi-scale visual prompting method with a planar concept, specifically designed for visual autoregressive (VAR) models. Through an analysis of information propagation, the authors introduce prompts within the outermost square frame. Combined with increasingly detailed tuning text, MVP enables efficient prompt learning at a low computational cost, enhancing the class-to-image capability of VAR and extending its text-to-image generation ability.

**Strengths:**

1. The paper proposes a perturbation-based visual prompt tuning method featuring a planar concept and an efficient information propagation mechanism, achieving competitive performance with relatively low training cost.
2. Through theoretical derivation and empirical analysis, the authors show that incorporating prompt features into the outermost frame of the feature map enables more efficient information propagation, and a scale threshold mechanism is introduced to balance performance and computational efficiency.
3. A multi-level semantic refinement strategy is employed during prompt training, enhancing the semantic representation and generation quality of VAR.

**Weaknesses:**

1. The paper lacks experimental comparisons with embedding-based methods.
2. Theoretical explanation is insufficient: although the authors analyze the advantages of introducing prompt features into the outermost frame of the feature map from the perspective of information propagation, the analysis lacks rigorous theoretical grounding. It would be helpful to include experimental comparisons of incorporating prompts at other positions within the feature map.
3. The experimental setup is constrained by the VAR framework, focusing mainly on the class-to-image task, which is neither the mainstream nor the most representative scenario for visual prompt tuning. Since class labels contain limited semantic information, this setup cannot fully demonstrate the potential of prompts in semantic alignment and generative control. Therefore, it remains unclear whether MVP can maintain its advantages in more typical text-conditioned or open-domain generation tasks.

**Questions:**

- Missing metrics: the paper does not report LPIPS or PSNR scores.

---

> ### Author Response · Authors · 2025-11-25
> **Part 1**
>
> Hi Reviewer `LEkF`, we sincerely appreciate you for taking the time to review our work and for providing both encouraging and constructive feedback. We highly value your comments and provide our detailed responses below.
>
> >**W1: Lack of experiments comparing with embedding-based methods.**
>
> Thank you for raising this question. However, we think that there may be a misunderstanding. The experiment “Multi-scale Prompt vs. Prefilled Prompt” (Table 4) already provides a comparison between MVP and embedding-based prompt. You may refer to the main text for more analysis details.
>
> &nbsp;
>
> >**W3: It is necessary to evaluate MVP on text-conditional or open-domain generation tasks to fully validate MVP's potential.**
>
> We sincerely thank the reviewer for raising this valuable point. We acknowledge the importance of evaluating whether MVP can generalize beyond class-conditional generation to text-conditional or more open-domain generation tasks, and we would like to clarify that our appendix already includes text-conditioned experiments based on two widely adopted VAR-like models, Infinity [r1] and HART [r2].
>
> To further demonstrate MVP’s applicability beyond class-conditional settings, we additionally provide an open-domain task experiment on VAR-based super-resilution task. Considering that HART adopts a more flexible rotary positional encoding than VAR and provides a stronger architectural backbone for spatially aligned prediction, we integrate MVP into the HART framework and compare it against VARSR[r3] as the baseline. This setup allows us to more fairly assess whether MVP can enhance an open-domain super-resolution task while avoiding limitations inherited from the original VAR architecture. The results evaluate under the DRealSR real-world image restoration benchmark are as follows:
>
> | **Model** | **PNSR ↑** | **SSIM ↑** | **LPIPS ↓** | **FID ↓** |
> |-----------|------------|----------| ----------|---------|
> | VARSR(baseline) | 28.16 | 0.7652 | 0.3541 | 155.87 |
> | HART + MVP (ours) | 27.67 | 0.7701 | 0.3626 | 151.38 |
>
> These results demonstrate that MVP is compatible with open-domain restoration pipelines and achieves competitive performance on such tasks, further supporting its ability to generalize beyond class-conditional generation.
>
> We believe that the above experiments sufficiently demonstrate MVP’s ability to extend to more open-domain tasks and effectively address the reviewer’s concerns.
>
> &nbsp;
>
> &nbsp;
>
> >**Q1: Missing metrics about LPIPS or PSNR scores.**
>
> We thank the reviewer pointing out the missing LPIPS and PSNR results. Following your suggestion, we computed both metrics for MVP and VAR under class-conditional ImageNet ($256 \times 256$). The results are as follows:
>
> | **Depth** | **Model** | **LPIPS ↓** | **PSNR ↑** |
> |-----------|-----------|-------------|------------|
> | 16        | MVP | 0.720 | 8.921 |
> | 16        | VAR | 0.738 | 8.844 |
> | 30        | MVP | 0.725 | 8.879 |
> | 30        | VAR | 0.711 | 8.732 |
>
> The results show that MVP and VAR obtain similarly low PSNR values and relatively high LPIPS distances.
>
> We thank the reviewer's suggestion but must clarify that these two metrics does not reflect the actual generative quality of the models. LPIPS and PSNR are designed for reconstruction-oriented tasks that assume pixel-aligned ground-truth targets (e.g., super-resolution or inpainting). In open-domain generative tasks class-conditional or text-conditional image generation, there is no unique reference image for each generated sample, and therefore these metrics inherently penalize all models regardless of architecture. This explains why MVP and VAR appear poor on LPIPS and PSNR despite strong FID/IS performance.
>
> &nbsp;
>
> &nbsp;
>
>
> Should you have further questions or wish to discuss any part of our work in more depth, we would be very glad to continue the conversation. Thank you!
>
> &nbsp;
>
> &nbsp;
>
>
> Best wishes and regards,
>
> All authors of Submission 769.
>
> &nbsp;
>
> &nbsp;
>
> &nbsp;
>
> &nbsp;
>
> **Reference**
>
> [r1] Infinity: Scaling bitwise autoregressive modeling for high-resolution image synthesis, CVPR2025
>
> [r2] Hart: Efficient visual generation with hybrid autoregressive transformer, ICLR 2025
>
> [r3] Visual autoregressive modeling for image super-resolution, ICML 2025

---

### Official Review · Reviewer_Lv5P · 2025-10-31

**Soundness:** 3
**Presentation:** 2
**Contribution:** 2
**Rating:** 2
**Confidence:** 5

**Summary:**

This paper introduces Multi-scale Visual Prompt (MVP), a novel parameter-efficient tuning (PEFT) method designed to enhance and expand the capabilities of Visual AutoRegressive (VAR) models for image generation. The authors argue that traditional autoregressive (AR) models struggle with a "planar concept" due to raster-scan order, which is addressed by adopting the VAR's next-scale prediction paradigm. MVP employs a perturbation-based prompt tuning strategy where learnable prompt tokens are introduced only at the outermost square frame of the feature map at each scale.

**Strengths:**

The work demonstrates a few promising aspects, particularly in efficiency and the choice of model:
* The paper explores perturbation-based visual prompt tuning in the relatively unexplored domain of visual autoregressive generation. The core idea of a multi-scale prompt restricted to the outermost frame is an original design choice for next-scale AR models
* The most notable strength is the computational efficiency achieved during tuning. MVP achieves T2I performance comparable to the fully trained VAR-CLIP while requiring only a fraction of the GPU-hours for training, highlighting its potential as an efficient fine-tuning mechanism for large VAR models.

**Weaknesses:**

* Lack of Rigorous Justification for Prompt Design (Planar Concept and Impact): The theoretical justification for selecting the outermost frame is weak and based on highly simplified models.
* The notion that selecting tokens in the outermost frame is sufficient to capture the "planar concept" based on the geometric principle of three non-collinear points is an oversimplification. It is unclear why three points are sufficient to control the plane of a high-dimensional feature map
* Insufficient Quantitative Validation on Text-to-Image (T2I) Task: While T2I generation is a key claimed contribution, the experimental validation is inadequate and does not use standard practices.
* The multi-scale nature and the multi-stage training are central to MVP, but the ablation studies (Tables 5, 6, 7) are too shallow.

**Questions:**

* Please provide a rigorous ablation study that compares the performance (FID/IS and efficiency) of the outermost square frame prompt against an equally-sized prompt budget (i.e., the same number of trainable tokens) placed at different strategic locations, such as a compact block in the center of the feature map, and a set of randomly distributed tokens. This is crucial to validate the claims about minimizing central impact and maximizing information propagation efficiency.
*  Provide an ablation to justify the complex, multi-stage tuning text strategy (labels $\rightarrow$ sentences $\rightarrow$ captions). Specifically, what is the performance difference when only the most detailed tuning text (captions) is used across all stages? Does the incremental complexity of the tuning strategy genuinely contribute to the final performance?

---

> ### Author Response · Authors · 2025-11-25
> **Part 1**
>
> Hi Reviewer `Lv5P`, we sincerely appreciate you for taking the time to review our work and for providing such constructive comments. We highly value your comments and provide our detailed responses below.
>
> >**W2: Why can a high-dimensional feature map be treated as having a planar concept ?**
>
> VAR’s next-scale prediction is essentially a multi-token prediction autoregressive modeling paradigm in which a set of tokens predicted at once is regarded as the next scale. The scale (i.e., feature map) inherently possesses a planar concept, constituting the very foundation upon which VAR operates, and it is also  widely acknowledged. While each token within a scale is high-dimensional, the token set as a whole is spatially arranged in two dimensions, allowing the entire feature map to be represented by a 2D plane. Based on this perspective, we respectfully argue that as long as at least three non-collinear tokens are selected as prompt tokens, our prompt design naturally preserves the planar concept. We hope this explanation clarifies our presentation and the consideration behind our design.
>
> &nbsp;
>
> >**W3:Insufficient Quantitative Validation on Text-to-Image (T2I) Task while T2I generation is a key claimed contribution.**
>
> We thank the reviewer for raising this point. However, we would like to respectfully clarify that we do not consider this aspect a weakness. The key claimed contribution of our work lies in the strong capability of MVP to extend and enhance VAR-based generative models. Tasks such as T2I and other conditioning settings are not claimed as core contributions. Instead, they serve as additional evidence of MVP’s robustness and versatility, illustrating that our prompt–placement design can generalize to multiple types of conditioning signals—even when applied to non-ideal or minimally engineered VAR backbones.
>
> More specifically, our motivation for T2I experiments is fundamentally different from constructing a competitive T2I model. Developing a full VAR-based T2I model requires substantial architectural redesign, a powerful language encoder, and large-scale training on high-quality T2I datasets—none of which aligns with the focus of our work. Rather, our experimental setup is intentionally designed to **explore the potential of MVP in extending VAR**, rather than to build a complete T2I system. So we intentionally adopt a lightweight and constrained setup similar to VAR-CLIP to evaluate whether MVP can meaningfully integrate textual information into VAR with minimal engineering and compute overhead. This positioning is further supported by our experiments: despite using less than 1% of the compute cost required by VAR-CLIP, MVP already enables VAR to perform basic yet effective T2I generation.
>
> Nevertheless, we also agree that a more comprehensive T2I evaluation  would further strengthen the results. To address this, we additionally provide more extensive T2I validation on VAR-like models based on HART, Infinity, and other stronger backbones in the appendix.

---

> ### Author Response · Authors · 2025-11-25
> **Part 2**
>
> >**Q2 & W4:Need more ablation studies for multi-scale nature and multi-stage training stage.**
>
> We sincerely thank you for this insightful comment. As discussed in Table A previous, the ablation on different prompt position designs already provides claer evidence for the multi-scale nuture of MVP. Following the reviewer's suggestion, we conduted an additional ablation to isolate the contribution of each textual tuning stage, we compared our defalt setting with three variants: (i) using only labels, (ii) using labels and sentences, (iii) using labels and captions. The results are as follows:
>
>
> **Table B. Ablation on prompt position designs under different token budgets. '*' indicates mean results over 3 runs with different random seeds**
>
> | **Depth** | **Tuning Stage** | **FID ↓** | **IS ↑** |
> |-----------|------------------|-----------|----------|
> | 16        | labels | 3.54 | 236.8 |
> | 16        | labels + sentences | 3.50 | 244.5 |
> | 16        | labels + captions | 3.48 | 242.8 |
> | 16        | labels + sentences + captions | 3.46 | 247.4 |
> | 30        | labels | 2.11 | 280.5 |
> | 30        | labels + sentences | 2.06 | 287.1 |
> | 30        | labels + captions | 2.06 | 291.3 |
> | 30        | labels + sentences + captions | 2.03 | 289.4 |
>
>
> The results show that our multi-stage prompt learning strategy are naturally with the coarse-to-fine generative process of VAR and yields the best overall FID/IS (3.46/247.4 at depth 16 and 2.03/289.4 at depth 30). We will move this ablation study into the main paper in the revised version, which will substantially improve the clarity and completeness of our experimental analysis.
>
> These ablation studies we provide further enrich the ablation analysis of the paper, and we believe they sufficiently address the reviewer’s concerns.
>
>
> &nbsp;
>
> &nbsp;
>
> &nbsp;
>
> &nbsp;
>
> Best wishes and regards,
>
> All authors of Submission 769.

---

### Author Response · Authors · 2025-12-02
**General Response(Part 2)**

>**2. `Lv5P` Q1 & W4 `LEkF` W2 `iowk` W1 `o6w3` W1 & Q3: Ablation of Other Prompt Design**.

We sincerely thank the reviewers for highlighting the need for a rigorous and complete ablation on alternative prompt position designs. Before presenting our extended results, we would like to respectfully clarify that this ablation was actually included in the initial submission; however, due to the main paper’s page limitations and our early assessment of its relative importance, we placed it in the appendix rather than in the main paper.

Following the reviewer's valuable suggestions and building on our previous experiments, we have further expanded this ablation by introducing three additional prompt-position design on the top of the original random-placement ablation in the appendix, all using an identical prompt token budget: (i) Innermost: Prompt tokens are first placed at the center of the feature map and then expand outward until the prompt token budget is reached; (ii) Center-to-Outer: Prompt tokens are placed on alternating concentric frames, starting from the innermost and skipping one frame each time; (iii) Sub-Outermost: Prompt tokens are placed from the second outermost square frame and extend outward if the prompt token budget is not yet filled.

We evaluate these designs on improving VAR’s class-conditional generation performance on ImageNet ($256 \times 256$), using FID and IS scores. The results are summarized as follows:

**Table A. Ablation on prompt position designs under equal token budget. '*' indicates mean results over 3 runs with different random seeds**

| **Depth** | **Placement Strategy**     | **FID ↓** | **IS ↑**  |
| --------- | -------------------------- | --------- | --------- |
| 16        | **Ours (Outermost-Frame)** | **3.46**  | **247.4** |
| 16        | Random                     | 3.69*     | 238.2*    |
| 16        | Innermost                  | 3.68      | 340.0     |
| 16        | Center-to-Outer            | 3.64      | 237.8     |
| 30        | **Ours (Outermost-Frame)** | **2.03**  | **289.4** |
| 16        | Sub-Outermost              | 3.51      | 243.1     |
| 30        | Random                     | 2.16*     | 281.7*    |
| 30        | Innermost                  | 2.13      | 283.3     |
| 30        | Center-to-Outer            | 2.14      | 279.6     |
| 30        | Sub-Outermost              | 2.08      | 285.2     |


These results reveal two key findings: (i) Prompt positions closer to the center consistently lead to worse performance, strongly validating the attenuation behavior we discussed;(ii) The outermost square frame strategy minimizes distributional distortion of the pretrained backbone while providing effective semantic guidance. We have moved this ablation study **from the appendix into the main paper** and **highlighted it in blue** for better visibility.

&nbsp;

&nbsp;

&nbsp;

&nbsp;

Best wishes and regards,
All authors of Submission 769.

&nbsp;

&nbsp;


Reference:

[r1] Paint-by-Example: Exemplar-based Image Editing with Diffusion Models, CVPR 2023

[r2] Drag Your GAN: Interactive Point-based Manipulation on the Generative Image Manifold, SIGGRAPH 2023

[r3] Boxdiff: Text-to-image synthesis with training-free box-constrained diffusion, CVPR 2023

[r4] Exploring the Transferability of Visual Prompting for Multimodal  Large Language Models, CVPR 2024

[r5] Exploring Visual Prompts for Adapting Large-Scale Models, arxiv 2022

[r6] Unleashing the Power of Visual Prompting At the Pixel Level, TMLR 2024

[r7] AutoVP: An Automated Visual Prompting Framework and Benchmark, ICLR 2024

---

### Author Response · Authors · 2025-12-02
**General Response(Part 1)**

We sincerely thank all reviewers and ACs for their time and efforts. We are encouraged that the reviewers recognize our motivation or originality (`Lv5P`,`o6w3`), consider our method novel or interesting (`Lv5P`,`o6w3`), and recognize the prospect and practicality of our method (`Lv5P`,`iowk`). In addition, all reviewers acknowledged that our method achieves competitive performance and computational efficiency.

In this work, we propose MVP, a perturbation-based prompt tuning method tailored for VAR and VAR-like models. This method is pretty simple yet highly effective, a characteristic that Reviewer `o6w3` explicitly recognized in review. MVP is the first perturbation-based prompt tuning method developed for VAR, and it is also the first method to introduce prompt learning at the multi-scale level. Further, MVP adapts to multiple downstream tasks. For the task of extending the VAR backbone from C2I to T2I under the setting that follows VAR-CLIP, MVP completes this task using only 0.54% of the training GPU hours while still achieving performance competitive with VAR-CLIP.

For the concerns that appeared consistently across the reviewers, we provide a unified clarification and response below:

>**`Lv5P` W1 & `LEKF` W2: More Theoretical Justification for Prompt Design**.

We sincerely thank the reviewers for the question about more theoretical justification. We would like to begin by acknowledging that the community currently lacks a solid theoretical foundation for this type of design. For instance, several recently published top-tier works [r1-r7] employ a similar strategy, i.e., perform operations or introduce control signals along the outer edge of the image or feature map, to ensure minimal impact on the image center. The image center typically contains key information and primary objects. Modifying the image center may lead to issues such as subject deformation, semantic drift, and expression distortion [r1-r3]. However, these works likewise do not provide theoretical analysis or justification. Therefore, the absence of a strong theoretical explanation is a limitation that extends across the field, rather than aunique to our work.

That being said, while adopting this design, we still hope to contribute to the field by offering deeper theoretical analysis. Motivated by this, we attempted to provide preliminary analysis, which led to the analysis presented in Section 3.1. However, to satisfy the reviewers, we have further expanded our analysis, aiming to deepen the theoretical analysis and provide clearer reasoning behind our design. Please refer to Section D.3 in the appendix.

The above represents our best efforts to provide analysis and justification for this design. However, we must reiterate that this represents the limitation of the entire field, and we have already pioneered the incorporation of theoretical analysis into this method.

Regarding the concern that our theoretical analysis is conducted on a simplified model, we respectfully view this as an advantage rather than a weakness. MVP is fundamentally built upon the next-scale prediction autoregressive paradigm, and VAR, the origin of next-scale prediction, serves as the conceptual basis for all subsequent variants and works. Conducting analysis on this simplified yet representative backbone is both sound and beneficial, as it enables clearer understanding of the core design and facilitates to generalize the our design to more advanced next-scale prediction models. Furthermore, our experiments show that applying MVP to advanced VAR-like models such as HART and Infinity indeed yields excellent performance (as shown in Appendix.B), which is the advantage of analyzing on the simplified model. This further demonstrates that our design is broadly effective within the next-scale prediction models and possesses strong generalization capability.

---

### Meta-Review · Area_Chair_2Vtw · 2026-01-07

**Summary:**

Initial scores were mixed. Most concerns focused on whether the design is scientifically well grounded. In particular, Lv5P, iowk, and o6w3 viewed the "planar concept" and "three points define a plane" claim as post hoc and not rigorous. They also asked for ablations that compare the "outermost" placement with other spatial layouts. They further pointed out the first-scale mismatch between C2I and T2I and the weak T2I results. The authors tried to answer these points, but the rebuttal still confirms the lack of a solid theory. The text-to-image extension, a key claim in original submission, is described by the authors as "exploratory" in rebuttal and not competitive with standard baselines. Overall, the paper does not meet the conference bar, and AC encourages the authors to fix these fundamental issues before submitting again to another venue.

**Reviewer Concerns:**

Addressed:

Prompt placement: The authors added ablations comparing "Outermost-Frame" with "Random", "Innermost", and "Center-to-Outer" placements.

Multi-stage training: They added an ablation showing label to sentence to caption training performs better than using captions alone.


Outstanding:

Theory: The rebuttal agrees the field lacks solid theory, so "three points define a plane" remains a heuristic.
First-scale mismatch: The approach still needs ad hoc tweaks, dropping prompts at the first scale for C2I but keeping them for T2I.

T2I extension: The T2I results are described as exploratory and not competitive, so the extension claim is weak.

**Reviewer Scores:**

Lv5P: Likely 2. The theory limitation remains weak.

iowk: Likely 4. The text-to-image part is still exploratory and not competitive.

o6w3: Likely 4. The first-scale inconsistency still looks ad hoc.

LEkF: Likely 6. Still positive overall.

---

### Decision · Program_Chairs · 2026-01-26

Reject